# Transcriptional regulation of autophagy-lysosomal function in BRAF-driven melanoma progression and chemoresistance

Shun Li[1,2], Ying Song[1], Christine Quach[1], Hongrui Guo[1,3], Gyu-Beom Jang[1], Hadi Maazi[1], Shihui Zhao[1], Nathaniel A. Sands[1], Qingsong Liu[4], Gino K. In[5], David Peng[6], Weiming Yuan[1], Keigo Machida[1], Min Yu[7], Omid Akbari[1], Ashley Hagiya[8], Yongfei Yang[1], Vasu Punj[8,9], Liling Tang[2] & Chengyu Liang[1]

Autophagy maintains homeostasis and is induced upon stress. Yet, its mechanistic interaction with oncogenic signaling remains elusive. Here, we show that in BRAF$^{V600E}$-melanoma, autophagy is induced by BRAF inhibitor (BRAFi), as part of a transcriptional program coordinating lysosome biogenesis/function, mediated by the TFEB transcription factor. TFEB is phosphorylated and thus inactivated by BRAF$^{V600E}$ via its downstream ERK independently of mTORC1. BRAFi disrupts TFEB phosphorylation, allowing its nuclear translocation, which is synergized by increased phosphorylation/inactivation of the ZKSCAN3 transcriptional repressor by JNK2/p38-MAPK. Blockade of BRAFi-induced transcriptional activation of autophagy-lysosomal function in melanoma xenografts causes enhanced tumor progression, EMT-transdifferentiation, metastatic dissemination, and chemoresistance, which is associated with elevated TGF-β levels and enhanced TGF-β signaling. Inhibition of TGF-β signaling restores tumor differentiation and drug responsiveness in melanoma cells. Thus, the "BRAF-TFEB-autophagy-lysosome" axis represents an intrinsic regulatory pathway in BRAF-mutant melanoma, coupling BRAF signaling with TGF-β signaling to drive tumor progression and chemoresistance.

[1] Department of Molecular Microbiology and Immunology, Keck School of Medicine, University of Southern California, Los Angeles, CA 90033, USA. [2] Key Laboratory of Biorheological Science and Technology, Ministry of Education, College of Bioengineering, Chongqing University, Chongqing 400044, China. [3] College of Veterinary Medicine, Sichuan Agriculture University, Chengdu 611130, China. [4] High Magnetic Field Laboratory, Chinese Academy of Sciences, 350 Shushan Hu Road, Hefei 230031, China. [5] Norris Comprehensive Cancer, Division of Oncology, University of Southern California, Los Angeles, CA 90033, USA. [6] Department of Dermatology, Keck School of Medicine, University of Southern California, Los Angeles, CA 90033, USA. [7] Department of Stem Cell Biology and Regenerative Medicine, Keck School of Medicine, University of Southern California, Los Angeles, CA 90033, USA. [8] Department of Pathology, Keck School of Medicine, University of Southern California, Los Angeles, CA 90033, USA. [9] Department of Medicine, Keck School of Medicine, University of Southern California, Los Angeles, CA 90033, USA. Correspondence and requests for materials should be addressed to L.T. (email: tangliling@cqu.edu.cn) or to C.L. (email: chengyu.liang@med.usc.edu)

Autophagy, originally described as a lysosome-dependent degradation of cytoplasmic components upon starvation, has since been shown to influence diverse aspects of homeostasis, constituting a barrier against malignant transformation[1]. Despite its inhibitory role in tumor initiation, autophagy is postulated to fuel the growth of established tumors and confers drug resistance, principally as a survival mechanism[1]. In melanoma, where 40–60% of cases have a mutation in BRAF, conflicting results have been reported regarding the relationship between autophagy and the BRAF[V600E] mutant, the most prevalent genetic alteration in melanoma[2]. On one hand, autophagy was found to overcome senescence and promote growth of BRAF[V600E]-driven melanoma in mice[3]. On the other, autophagy was shown to suppress BRAF[V600E]-driven tumorigenesis, and reduced expression of autophagy-related *Atg* genes was observed in melanoma patients[4]. Despite the ambiguous interaction between BRAF signaling and autophagy, autophagy was consistently induced in melanoma patients who were given highly specific BRAF[V600E] inhibitors (BRAFi)[5]. Several mechanisms for BRAFi-induced autophagy have been proposed, involving activation of ER stress or AMP-activated protein kinase[6,7]. None of them, however, explain the intrinsic link between BRAF signaling and autophagy. Thus, a better understanding of the interaction between autophagy and tumor growth control is necessary to improve cancer treatments.

Although autophagy functions through the orchestrated actions of *Atg* gene products in the cytoplasm, the control center resides in the nucleus, whereby the microphthalmia/transcription factor E (MiT/TFE) transcription factors, particularly transcription factor EB (TFEB) and transcription factor E3 (TFE3), regulates most *Atg* gene expression in coordination with the genes involved in lysosomal biogenesis/function[8]. Elevated autophagy–lysosomal function is the direct consequence of TFEB/TFE3 activation[8,9]. Current studies indicate that TFEB/TFE3 are regulated by mammalian target of rapamycin complex 1 (mTORC1)[8]. Under basal conditions, TFEB/TFE3 are phosphorylated by mTORC1 at S142 or S211 in TFEB or S321 in TFE3[10,11]. TFEB/TFE3 phosphorylation creates docking sites for the 14-3-3 proteins, causing cytoplasmic sequestration of TFEB/TFE3 as an off-state[8]. Starvation/lysosomal stress releases mTORC1 from the lysosome, and consequently, non-phosphorylated TFEB/TFE3 translocate to the nucleus and induces expression of autophagy–lysosome-relevant genes[8,12]. Notably, extracellular signal–regulated kinase (ERK) is also shown to phosphorylate TFEB at S142 and regulate its nuclear translocation;[12] yet, the significance of this regulation by ERK vs. that by mTORC1 remains uncertain. Furthermore, zinc finger with KRAB and SCAN domains 3 (ZKSCAN3)[13], a transcriptional repressor of the autophagy–lysosome network, is regulated in conjunction with TFEB during starvation/lysosome activation through c-Jun N-terminal kinase 2/p38 mitogen-activated protein kinase (JNK2/p38 MAPK)-mediated phosphorylation[14]. The orchestrated regulation of the autophagy–lysosomal system by TFEB/ZKSCAN3 highlight the importance of this pathway in cellular adaptation to environmental cues, which might be altered in pathological settings such as cancer.

Despite advanced knowledge of the autophagy–lysosomal regulation during stress, the precise mechanism by which this pathway responds to oncogenic signaling remains unclear. Here, we identify the molecular basis by which BRAF[V600E] controls the transcriptional machinery of the autophagy–lysosomal pathway through TFEB in melanoma. Constitutive TFEB phosphorylation by the BRAF[V600E] downstream effector ERK leads to its cytoplasmic retention and impaired expression of autophagy–lysosome target genes, which can be reversed by BRAFi. In conjunction with TFEB activation, BRAFi increases JNK2/p38-mediated phosphorylation/

inactivation of ZKSCAN3. Blockade of BRAFi-induced autophagy–lysosomal activation in BRAF-mutant melanoma causes increased tumor progression, epithelial-to-mesenchymal-like transition (EMT), and partial resistance to BRAFi therapy. Furthermore, we identified transforming growth factor-β (TGF-β) signaling as a key pathway downstream of TFEB inactivation. Inhibition of TGF-β signaling reverted EMT and restored BRAFi responsiveness in BRAF-mutant melanoma. These findings delineate a mechanism by which BRAF[V600E] regulates TFEB to reshape the autophagy–lysosomal framework in melanoma growth.

## Results

**BRAFi promotes autophagy–lysosome biogenesis through TFEB.** To investigate how oncogenic BRAF regulates autophagy in melanoma, we treated A375 human melanoma cells, which express BRAF[V600E], with PLX4720, a selective BRAFi[15]. We determined the subcellular distribution of the autophagy marker GFP-LC3, and the levels of lipidated LC3 (LC3-II) and of p62, an autophagic substrate. In agreement with previous findings[6], the levels of LC3-II production, GFP-LC3 puncta, and p62 degradation were increased in PLX4720-treated A375 cells in a dose-dependent manner, as seen with starvation (Fig. 1a, b and Supplementary Fig. 1a). In contrast, autophagy was not induced in wild-type (WT) BRAF-containing MeWo cells in response to PLX4720 (Fig. 1a, b and Supplementary Fig. 1a). Besides autophagy, PLX4720-induced lysosomal expansion, as indicated by LysoTracker and LAMP1 staining, and increased lysosomal protease activities, as measured by β-N-acetylglucosaminidase (NAG) assays, in A375, but not in MeWo cells (Fig. 1c, d and Supplementary Fig. 1b). Thus, BRAFi elicits concurrent activation of both autophagy and lysosome biogenesis/function.

To understand how PLX4720 activates the autophagy–lysosomal pathway, we performed quantitative PCR (qPCR) using mRNA extracted from PLX4720-treated A375 cells. PLX4720 increased expression of most autophagy–lysosome-related genes (Fig. 1e and Supplementary Fig. 1c). Furthermore, gene set enrichment analysis (GSEA) of multiple independent datasets[16–18] revealed that BRAFi treatment of BRAF-mutant melanomas caused a significant increase in the expression of the autophagy–lysosome gene set (Supplementary Fig. 1d). The autophagy–lysosomal pathway is transcriptionally controlled by the MiT/TFE factors such as MITF, TFE3, and TFEB[12]. Knockdown (KD) by independent short hairpin RNA (shRNA) for TFEB, but not for MITF or TFE3, abolished PLX4720-induced autophagy–lysosomal activation, whereas KD of endoplasmic reticulum (ER) stress-signaling molecules such as Glucose-regulated protein 78/Binding immunoglobulin protein (GRP78/BiP) or protein kinase R (PKR)-like endoplasmic reticulum kinase (PERK), previously implicated in BRAFi-induced autophagy[6], had minimal effects (Figs. 1f–i and Supplementary Fig. 1e–h). These findings were further confirmed by gene expression analysis, whereby PLX4720-associated upregulation of the autophagy–lysosomal signature was reduced upon TFEB KD (Fig. 1e and Supplementary Fig. 1c). Thus, TFEB is primarily responsible for BRAFi-associated transcriptional activation of autophagy and lysosomal function in melanomas.

**TFEB is activated by BRAFi via ERK inhibition.** PLX4720 treatment of A375 cells triggered nuclear translocation of TFEB, but not that of TFE3, whereas MITF was located in the nucleus regardless of treatment (Fig. 2a, b). Similar results were obtained in other BRAF[V600E]-positive melanoma cells such as G361 and SK-MEL-5, but not in MeWo and NRAS mutant SK-MEL-2 cells that contain WT BRAF (Supplementary Fig. 2a–d). Notably, BRAF[V600E]-positive HT29 colon cancer cells also exhibited TFEB

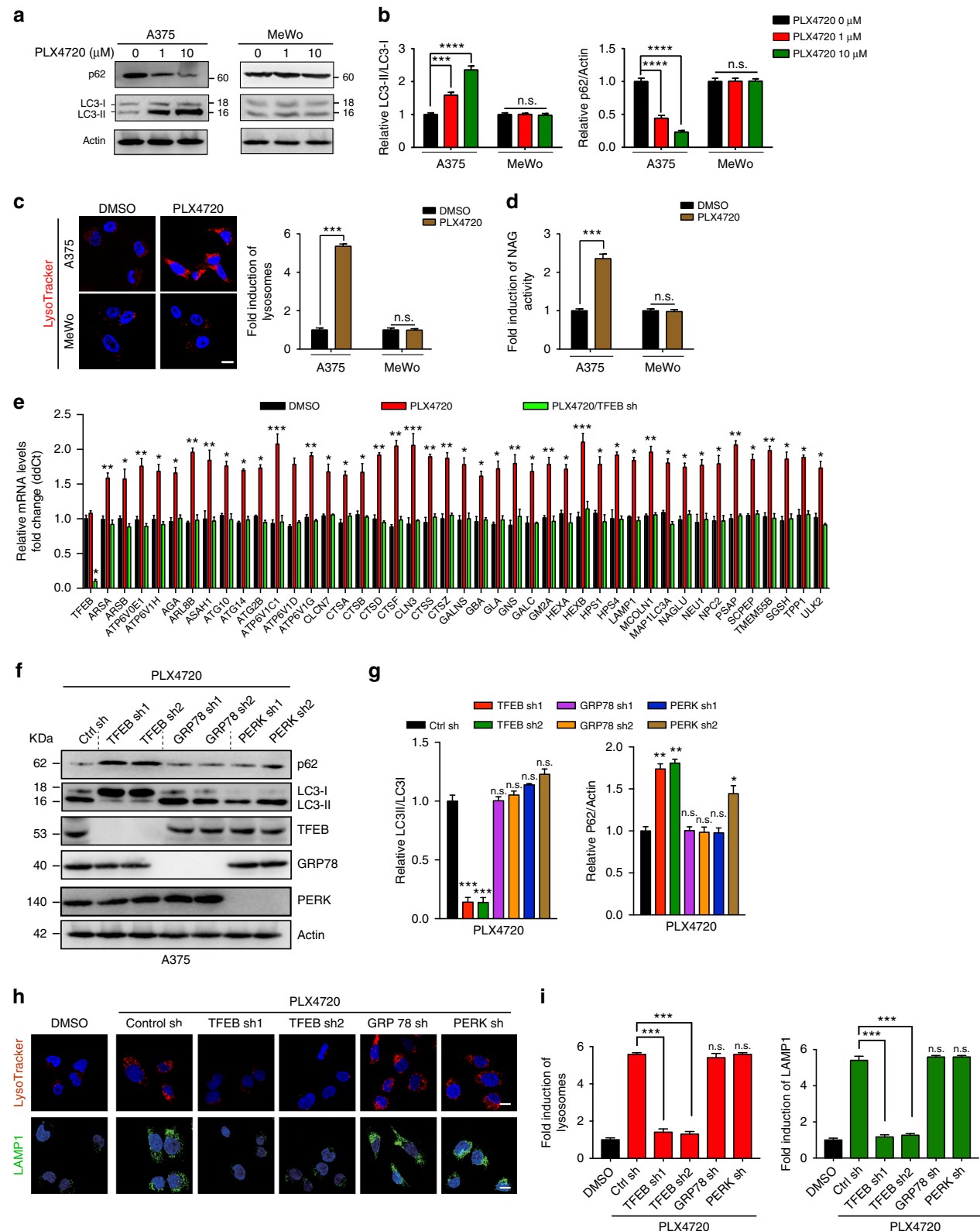

nuclear translocation upon PLX4720 treatment, suggesting a BRAF^V600E-specific event (Supplementary Fig. 2d). Cell fractionation analysis confirmed the nuclear enrichment of TFEB, but not MITF nor TFE3, induced by PLX4720, which was associated with a faster-migrating form of TFEB (Fig. 2c), suggestive of reduced phosphorylation[8]. Indeed, PLX4720 reduced the phosphorylation of the 14-3-3-binding motif (S211) in overexpressed

and endogenous TFEB without affecting their expression, and abolished 14-3-3/TFEB complex formation in A375 and G361, but not in MeWo cells (Fig. 2d, e and Supplementary Fig. 2e–g). These data suggest that BRAFi decreases phosphorylation, nuclear translocation, and thus activates TFEB.

TFEB is known to be phosphorylated and inhibited by mTORC1 in resting cells;[8] therefore, we examined whether

**Fig. 1** BRAF[V600E] inhibitor triggers autophagy–lysosomal activation through TFEB. **a**, **b** Western blot analysis (**a**) and densitometric quantification (**b**) of the LC3-II/LC3-I and the p62/Actin ratios in A375 and MeWo cells treated with the indicated concentrations of PLX4720 for 12 h. Actin served as a loading control. n = 4 independent experiments. **c** Representative images of LysoTracker Red staining of A375 and MeWo cells treated for 12 h with DMSO or PLX4720 (1 μM). Quantification of relative fold induction of lysosomes by PLX4720 is shown in the right panel. n = 3 independent experiments. **d** Relative lysosome NAG activity in PLX4720-treated A375 cells. n = 3 independent experiments. **e** Expression analysis of the autophagy–lysosome relevant genes in PLX4720-treated A375 cells in the presence or absence of TFEB (shRNA). n = 3 independent experiments. **f**, **g** Western blot analysis (**f**) and densitometric quantification (**g**) of the LC3-II/LC3-I and p62/Actin ratios in PLX4720-treated A375 cells with shRNA-mediated depletion of the indicated genes. Expression of indicated proteins is also shown. Actin served as a loading control. n = 3 independent experiments. **h**, **i** Representative images (**h**) and quantification (**i**) of LysoTracker Red (red) and LAMP1 (green) immunostaining of PLX4720 (1 μM, 12 h-treated) A375 cells with depletion of the indicated genes. Note the reduced lysosome staining in PLX4720-treated cells upon TFEB depletion. n = 3 independent experiments. Scale bars, 10 μm. Data in **a** and **f** are from one experiment that is representative of three independent experiments. For all quantification, data represent the mean ± SD derived from the indicated number of independent experiments. Comparisons were made using Student's t-test. *P < 0.05; **P < 0.01; ***P < 0.001; ****P < 0.0001; n.s. not significant. See Supplementary Fig. 13 for uncropped data of **a**, **f**

PLX4720 regulates TFEB through mTORC1. PLX4720 treatment resulted in decreased phosphorylation of p70S6K and 4E-BP1, two known mTOR substrates, suggesting alteration of mTORC1 signaling (Supplementary Fig. 2h). However, constitutive mTORC1 activation through ectopic expression of constitutively active (CA) mTORC1 (E2419K)[19] or RagB (Q99L) GTPase upstream of mTORC1[20], or depletion of DEPDC5, a key subunit of the GTPase-activating proteins toward Rags 1 complex[21], failed to preclude PLX4720-driven TFEB nuclear translocation in A375 cells (Fig. 2f), arguing against a critical role for mTORC1 in PLX4720-associated TFEB activation. We also examined other kinases or factors previously implicated in TFEB regulation such as GSK3, Akt, JNK1/2, p38 MAPK, IPO8, MCOLN1, and CRM1[14,22–25]. PLX4720 did not appreciably affect the activity of GSK3α/3β, as measured by its phosphorylation at Ser21 or Ser9, respectively[26], nor the pathways associated with Akt activation, as indicated by Ser473 and Thr308 phosphorylation[27] (Supplementary Fig. 2h). There was an apparent increase in the phosphorylation of JNK1/2 and p38 MAPK, reflecting their activation; yet, KD of JNK1/2 or p38 MAPK had no effect on TFEB nuclear translocation induced by PLX4720 (Supplementary Fig. 2h–k). Similarly, removal of MCOLN1, which activates calcineurin to dephosphorylate TFEB in response to lysosomal calcium signaling[24], IPO8, which regulates TFEB nuclear transport[23], or CRM1, which regulates TFEB nuclear export[25], or addition of leptomycin B (LMB), a known CRM1 inhibitor, did not alter TFEB localization in PLX4720-treated A375 cells (Supplementary Fig. 2h–k).

As PLX4720 inhibited ERK (Supplementary Fig. 2h), we examined the role of ERK in PLX4720-associated TFEB regulation. Indeed, expression of the CA ERK (R67S/D321N)[28] rendered TFEB insensitive to PLX4720, preventing its nuclear translocation (Fig. 2f). Consistently, treatment with the ERK inhibitor FR180204[29] or depletion of ERK by two different shRNA, was sufficient to elicit TFEB nuclear translocation and enrichment in A375 cells, but not in MeWo cells, mimicking the effect of PLX4720 (Fig. 2f–h). In contrast, TFE3 and MITF remained unaffected as noted before (Fig. 2h). Moreover, inhibition of p90S6K downstream of ERK had no effect on TFEB localization (Supplementary Fig. 2l). These data demonstrate that ERK is mainly responsible for PLX4720-induced TFEB cytoplasm-to-nucleus translocation in BRAF[V600E] melanoma cells.

**mTORC1-independent TFEB modification by ERK**. To understand the role of ERK in TFEB regulation in BRAF[V600E] melanoma, we performed co-immunoprecipitation (co-IP) and found that more TFEB co-IP with ERK in A375 than in MeWo cells, and that this association was disrupted by PLX4720 in

A375 cells (Fig. 3a). Moreover, colocalization of TFEB and ERK was observed in a juxtanuclear pattern co-stained with LAMP1 (Fig. 3b). PLX4720 dissociated ERK from the lysosomes, but not mTOR, and concomitantly induced TFEB nuclear translocation (Fig. 3b and Supplementary Fig. 3a), suggesting that TFEB phosphorylation by ERK likely occurs at the lysosomes. To examine whether TFEB lysosomal association is due to ERK-mediated phosphorylation, we constructed TFEB mutants in which the three putative ERK phosphorylation sites, i.e., T50, S389, and the previously identified S142[12], were mutated to alanine either individually or in double/triple combinations (Supplementary Fig. 3b). Unlike T50A or S389A, expression of the S142A mutant was sufficient to cause TFEB nuclear translocation. No additive effect was observed when S142A was combined with T50A and/or S389A (Fig. 3c). Using a S142 phospho-specific antibody, we found that TFEB S142 phosphorylation was enriched in A375 compared with MeWo cells (Fig. 3a). PLX4720 ablated TFEB S142 phosphorylation in A375 cells (Fig. 3a). An in vitro kinase assay using bacterially purified proteins showed that active ERK directly phosphorylates TFEB, and that was abolished by S142A, but not by S211A, a TFEB site reported to be phosphorylated by mTORC1[30] (Fig. 3d). Consistently, overexpression of ERK(CA) failed to block nuclear accumulation of TFEB[S142A], but blocked PLX4720-induced and S211A-induced nuclear translocation (Fig. 3e). Moreover, silencing CRM1 or treating cells with LMB failed to accumulate TFEB (WT and S211A) in the nucleus upon ERK(CA) expression, again highlighting a CRM1-independent regulation (Fig. 3e). Although S142 was also found to be phosphorylated by mTORC1[10], mTORC1 inhibition by depletion of Raptor, a key subunit of the mTORC1 complex, could not reverse the effect of ERK(CA) on the cytoplasmic retention of WT and TFEB[S211A] (Fig. 3e). In addition, TFEB[S142E] remained associated with the lysosomes and insensitive to both PLX4720 and mTORC1 inhibition by Raptor depletion (Fig. 3f). Thus, ERK-mediated S142 phosphorylation is required for TFEB lysosomal association that is disrupted by PLX4720 in an mTORC1-independent manner.

We further examined whether ERK regulates TFEB cytoplasmic retention by 14-3-3. The S142A mutant strongly reduced phosphorylation of the 14-3-3-binding motif in TFEB, as seen with S211A, and consequently TFEB-14-3-3 complex formation (Fig. 3g, h). In contrast, the TFEB[S142E] mutant escaped PLX4720 regulation and remained cytoplasmic, even in the presence of S211A that was previously noted to be sufficient for TFEB translocation (Supplementary Fig. 3c). The TFEB[S142E] mutant, and the TFEB[S142E/S211A] mutant to a lesser extent, interacted with 14-3-3, as did WT (Fig. 3h). Conversely, the presence of S211E could not inhibit the S142A-induced dissociation from 14-3-3 and nuclear translocation of TFEB (Fig. 3h and Supplementary

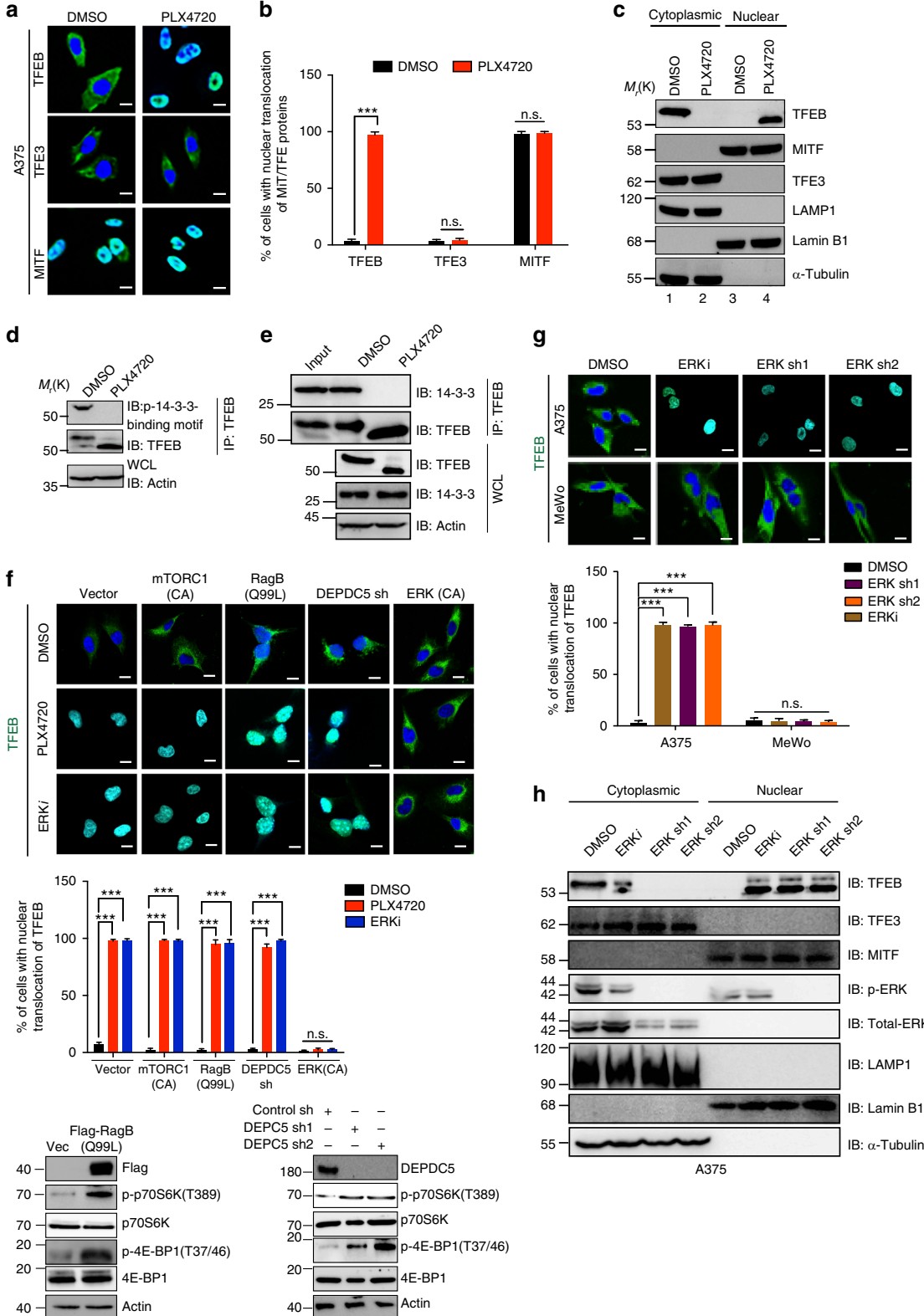

Fig. 3c). Thus, S142 phosphorylation by ERK plays a dominant role over S211 phosphorylation in TFEB regulation in BRAF-mutant melanoma.

Next, we tested the relative impact of S142 versus S211 on the transcriptional output of TFEB in response to PLX4720. TFEB$^{S142A}$ increased the basal levels of autophagy and lysosome biogenesis, including in the co-existence of S211E, whereas TFEB$^{S142E}$ suppressed autophagy and lysosome biogenesis, even in the presence of PLX4720 (Fig. 3i–k and Supplementary Fig. 3d). These findings imply that ERK-mediated TFEB S142 phosphorylation constitutes the dominant mechanism underlying PLX4720-induced autophagy–lysosomal activation in melanoma, and that BRAF$^{V600E}$ is a negative regulator of TFEB-dependent transcription.

**Fig. 2** PLX4720 promotes TFEB activation through ERK inhibition. **a** Confocal analyses of the subcellular distribution of endogenous TFEB, TFE3, and MITF in A375 cells treated with DMSO or PLX4720 (1 μM, 12 h). Nuclei were stained with DAPI (blue). $n = 3$ independent experiments. **b** Quantification of nuclear translocation of TFEB, TFE3, and MITF in cells in (**a**). $n = 150$ cells, pooled from three independent experiments. **c** Immunoblots for TFEB, MITF, and TFEB3 in the cytoplasmic/nuclear fractions of A375 cells treated with PLX4720 (1 μM, 12 h). Lamin B1 is the control for the nuclear fractions, whereas LAMP1 and Tubulin are the controls for the cytoplasmic fractions. **d** Immunoblotting of endogenous TFEB and p-14-3-3-binding motif of TFEB from PLX4720 (1 μM, 12 h-treated) A375 cells. WCL whole-cell lysate. **e** PLX4720 disrupts TFEB interaction with 14-3-3. WCLs of A375 cells were immunoprecipitated (IP) with anti-TFEB, followed by immunoblotting (IB) with antibodies against 14-3-3 and TFEB. **f** Representative images (top) and quantification (middle) of nuclear translocation of TFEB in A375 cells stably expressing mTORC1 (E2419K), RagB GTPase (Q99L), or ERK (R67S/D321N), or DEPDC5-specific shRNA, with or without the treatment of PLX4720 (1 μM, 12 h), or ERK inhibitor FR180204 (ERKi, 10 μM, 24 h). IB showed protein expression as indicated with the corresponding mTORC1 activity (p-p70S6K and p-4E-BP1). $n = 4$ independent experiments. **g** Representative confocal images (top) and quantification (bottom) of nuclear localization of endogenous TFEB in A375 and MeWo cells treated with FR180204 or with ERK shRNA. $n = 4$ independent experiments. **h** Immunoblots for endogenous TFEB, TFE3, MITF, and ERK in cytoplasmic/nuclear fractions of A375 cells treated with DMSO or FR180204 (10 μM, 24 h) or with ERK shRNA. Scale bars, 10 μm. Data in **c**, **d**, **e**, **f**, and **h** are from one experiment that is representative of three independent experiments. For all quantification, data represent the mean ± SD derived from the indicated number of independent experiments. Comparisons were made using Student's $t$-test. ***$P < 0.001$; n.s. not significant. See Supplementary Fig. 13 for uncropped data of **c**, **e**, **f**, **h**

**Coordinated ZKSCAN3 translocation and inactivation by BRAFi.** Given that PLX4720-treated A375 exhibited increased p38 MAPK and JNK activation (Supplementary Fig. 2h), we wondered whether BRAFi regulates ZKSCAN3 in conjunction with TFEB. Indeed, PLX4720 induced the rapid cytoplasmic translocation and enrichment of ZKSCAN3 concurrently with TFEB nuclear translocation in A375 and G361, but not in MeWo cells (Fig. 4a–c; Supplementary Figs. 4a, b). Translocation of ZKSCAN3 was blocked by depletion of p38 MAPK or JNK2, but not JNK1 or PKCδ, a known upstream regulator of ZKSCAN3[14] (Fig. 4d–f), whereas nuclear translocation of TFEB was not affected. Furthermore, overexpression of ERK(CA) or mTORC1 (CA) had minimal effects on ZKSCAN3, as it remained cytoplasmic following PLX4720 treatment, suggesting that ZKSCAN3 translocation is mTOR- and ERK independent but JNK2/p38 dependent (Fig. 4d–f). Notably, preventing ZKSCAN3 phosphorylation by mutating T153, a target site of JNK2/p38 MAPK upon PKCδ activation[14], abolished PLX4720-induced ZKSCAN3 translocation (Supplementary Fig. 4c), highlighting a conserved regulation of ZKSCAN3 in BRAF[V600E] melanoma. Consistent with its function as a transcriptional repressor[13], overexpression of ZKSCAN3[T153A], but not WT, abolished the effect of PLX4720 on autophagy–lysosome regulation (Supplementary Fig. 4d–f). Our results are consistent with a model where BRAF[V600E] inhibition couples the reduced phosphorylation, nuclear translocation, and activation of TFEB with increased phosphorylation, cytoplasmic relocation, and thus inactivation of ZKSCAN3 through ERK- and JNK2/p38 MAPK-dependent mechanisms, respectively, which results in increased net production of autophagy–lysosome-relevant factors (Fig. 4g).

**TFEB/ZKSCAN3 regulation mechanism in melanoma xenograft.** To assess whether BRAF[V600E]-driven melanoma progression requires TFEB S142 phosphorylation, we xenografted NOD/SCID mice with A375 tumors and found that, upon expression of non-phosphorylatable TFEB[S142A], or to a lesser extent WT, BRAF[V600E]-dependent tumor growth was inhibited and tumors exhibited increased autophagy (p62 and LC3-II) and lysosomal (Cathepsin D) functions compared with controls (Fig. 5a–c and Supplementary Fig. 5a). In contrast, accelerated tumor growth and suppressed autophagy–lysosomal function was observed in A375 xenografts that expressed the phosphomimetic TFEB[S142E] (Fig. 5a–c and Supplementary Fig. 5a). In agreement, the colony-forming ability of A375 cells was enhanced by TFEB[S142E], but impaired by TFEB[S142A] (Supplementary Fig. 5b). To corroborate this, we generated TFEB[KD] A375 cells and reconstituted them with WT/mutant TFEB to physiological levels, and confirmed that TFEB[S142E] promoted clonogenicity and xenograft growth,

whereas TFEB[S142A] reduced it (Supplementary Fig. 5c–e). Further, ZKSCAN3 depletion synergized with TFEB overexpression and resulted in growth inhibition of A375 xenograft melanoma (Supplementary Fig. 5f–h). These results indicate that TFEB/ZKSCAN3-dependent regulation of the autophagy–lysosomal pathway suppresses BRAF[V600E] melanoma progression. Inactivation of TFEB by ERK-mediated phosphorylation may therefore contribute to the oncogenic properties of BRAF[V600E].

Immunohistochemical analyses confirmed increased staining of Ki67[+] (proliferating) in A375 xenografts expressing TFEB[S142E], which was associated with increased apoptosis as indicated by active caspase-3 staining and by induction of PARP (Fig. 5d and Supplementary Fig. 5a). Electron microscopy (EM) analyses showed that TFEB[S142E] and vector-expressing tumors contained fewer numbers of mitochondria, and greater numbers of damaged ones with cristae loss, whereas only a few swollen mitochondria were in WT or TFEB[S142A] tumors (Fig. 5e and Supplementary Fig. 5i). Accumulation of damaged mitochondria correlated with enhanced ROS production, as indicated by 4-Hydroxynonenal (4-HNE) staining, and increased oxidative stress, as indicated by 8-OXO-dG (Fig. 5d). As such, TFEB[S142E] tumors exhibited increased genomic instability, as indicated by high levels of γ-H2AX[31] (Supplementary Fig. 5a) and necrosis (Fig. 5d), marked by the area of cells with pyknotic nuclei, karyorrhexis, and eosinophilic cytoplasm[32]. In addition, TFEB[S142E] tumors were poorly differentiated with nuclear pleomorphism and increased nucleus–cytoplasmic ratios, and highly invasive (Fig. 5d). In fact, immunoblots (Fig. 5f) and qRT-PCR analyses (Supplementary Fig. 6a) for the markers of EMT demonstrated a partial EMT-like process in TFEB[S142E] tumors, which was reversed in TFEB[S142A] tumors. Although melanocytes do not belong to the epithelial lineage, E-cadherin is required for melanocyte differentiation and suppresses their proliferation; loss of E-cadherin is associated with tumor progression and metastasis of melanoma[33]. These results indicate that TFEB S142 phosphorylation and the resultant suppression of autophagy–lysosomal transcription serve as downstream effectors of BRAF[V600E], contributing to tumor progression and poor differentiation.

**TFEB S142 phosphorylation activates TGF-β signaling.** To uncover the signaling pathways altered by TFEB phosphorylation, we performed unbiased RNA-seq of A375 melanoma xenografts. Clustering of the individual TFEB gene expression profiles generated two distinct clusters of gene expression signatures in TFEB WT/S142A and Vector/S142E tumors (Supplementary Fig. 6b). Most of the autophagy–lysosome-related genes were upregulated by TFEB[S142A], but downregulated by TFEB[S142E] (Fig. 5g and Supplementary Data 1a), consistent with the notion that the

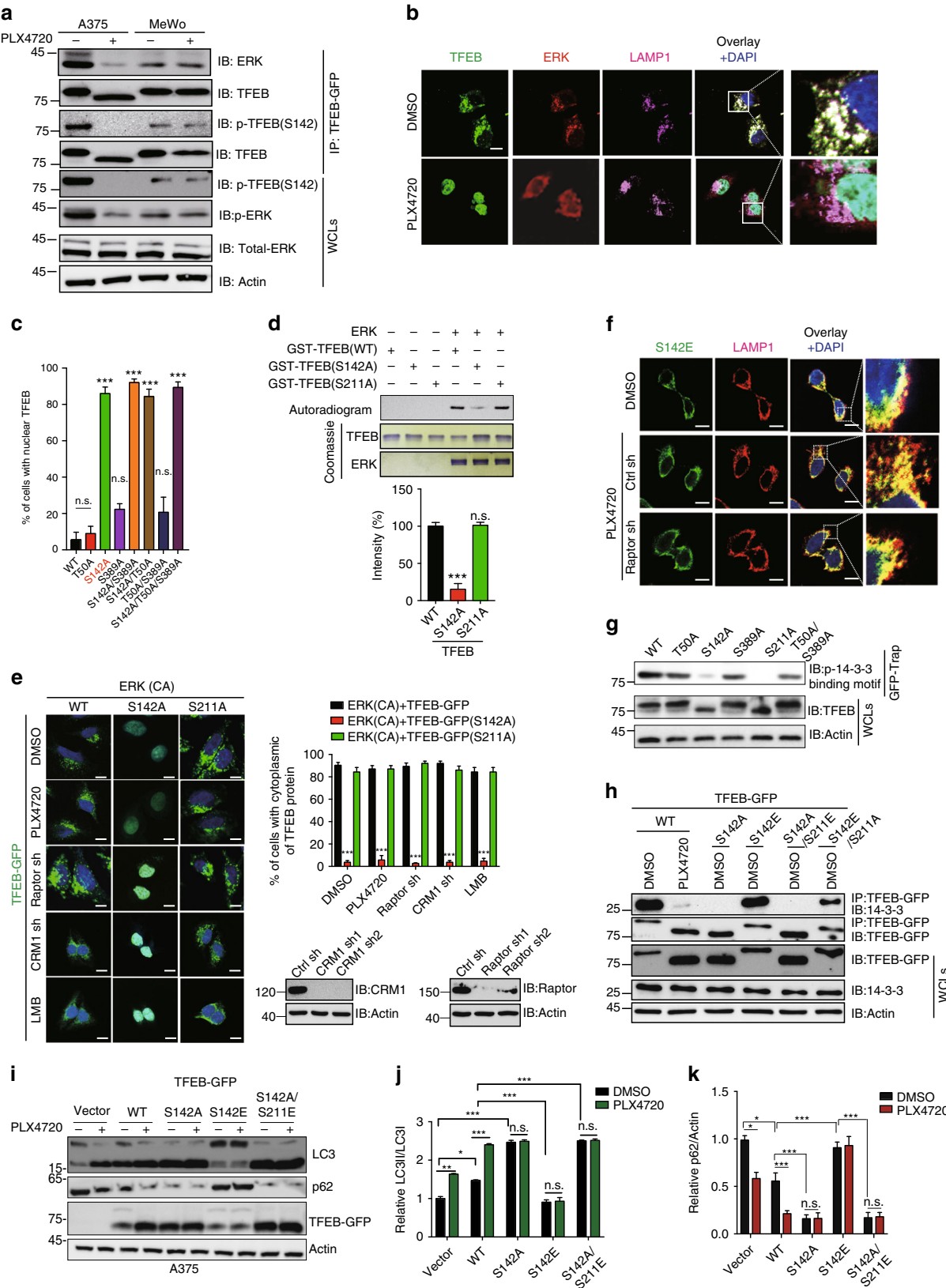

autophagy–lysosomal pathway is a major transcriptional output of TFEB[8]. Increased autophagy–lysosomal signatures also correlated with the upregulation of genes regulating melanocyte differentiation (Supplementary Fig. 6c; Supplementary Data 1b). On the contrary, the signatures that were upregulated by TFEB[S142E], but downregulated by TFEB[S142A], were related to EMT (Supplementary Fig. 6d and Supplementary Data 1c) and TGF-β target genes (Fig. 5h; Supplementary Fig. 6e; and Supplementary Data 1d–f). TGF-β signaling is a direct mediator of EMT, promoting expression of EMT-inducing transcription factors

**Fig. 3** TFEB S142 phosphorylation in BRAF$^{V600E}$ melanoma cells. **a** PLX4720 inhibits TFEB-ERK interaction and TFEB S142 phosphorylation. A375 and MeWo cells expressing TFEB-GFP were treated with PLX4720 (1 μM, 12 h), and WCLs were IP with anti-GFP-Trap beads, followed by IB with the indicated antibodies. $n = 3$. **b** PLX4720 releases ERK and TFEB from the lysosomes. $n = 3$. **c** Quantification of nuclear localization of TFEB-GFP in A375 cells expressing indicated TFEB mutants. $n = 3$. **d** In vitro phosphorylation of WT or TFEB mutants by ERK (CA). Phosphorylated TFEB were detected by autoradiography (top). The same gel was stained with Coomassie Blue to visualize total proteins (middle). Relative fold change in TFEB phosphorylation was quantified (bottom). **e** Representative images and quantification of cytoplasmic retention of WT, S142A, and S211A TFEB in A375 cells expressing ERK(CA) w/ or w/o treatment of PLX4720, Leptomycin B (LMB; 20 nM, 2 h), Raptor- or CRM1-shRNA. $n = 3$. **f** TFEB$^{S142E}$ associates with the lysosomes regardless of the treatment of PLX4720 or Raptor-shRNA. $n = 3$. **g** S142A mutation reduces phosphorylation of the 14-3-3-binding motif in TFEB. WT or mutant TFEB-GFP was expressed in A375 cells and IP with GFP-Trap beads, followed by IB with the indicated antibody. $n = 3$. **h** The predominant role of S142 phosphorylation over S211 phosphorylation in TFEB interaction with 14-3-3. A375 cells as indicated were treated with PLX4720, followed by IP with GFP-Trap beads and IB for the indicated proteins. $n = 3$. **i–k** IB (**i**) and quantification of the LC3-II/LC3-I (**j**) and p62/Actin ratios (**k**) in A375 cells expressing WT/mutant TFEB w/ or w/o PLX4720 treatment. $n = 3$. Scale bars, 10 μm. Data in **a**, **d**, **g**, **h**, and **i** are from one that is representative of three independent experiments. For all quantification, data represent the mean ± SD derived from the indicated number of independent experiments. Comparisons were made using Student's t-test. *$P < 0.05$; **$P < 0.01$; ***$P < 0.001$; n.s. not significant. See Supplementary Fig. 13 for uncropped data of **a**, **e,h**, **i**

(EMT-TFs) such as Twist1, ZEB1, Snail, and Slug[34]. Upregulation of these EMT-related TGF-β target genes was confirmed in TFEB$^{S142E}$-expressing A375 xenografts by qRT-PCR (Supplementary Fig. 6a) and immunoblot analyses (Fig. 5f). This suggests that TFEB S142 phosphorylation results in transcriptional induction of EMT, probably through TGF-β pathway, and that TFEB$^{S142A}$ suppresses TGF-β signaling.

To understand how TFEB regulates TGF-β signaling, we investigated the effect of TFEB S142 phosphorylation on key components of the TGF-β pathway. TFEB$^{S142A}$ strongly decreased, whereas TFEB$^{S142E}$ increased, levels of both latent and active TGF-β in A375 xenografts, but TGF-β receptor II (TGF-βRII) levels remained unchanged across xenografts (Fig. 5f). As a result of elevated TGF-β production, Smad2/3, the key mediators of TGF-β signaling, were activated as indicated by a strong increase in their phosphorylation in TFEB$^{S142E}$ tumors (Fig. 5f). Blockade of TGF-β signaling by a small-molecule inhibitor (TGF-βi) downregulated TFEB$^{S142E}$-associated EMT-TFs upregulation and EMT, whereas recombinant TGF-β promoted EMT, which was suppressed by TFEB$^{S142A}$ (Fig. 5i). These results show that elevated TGF-β and activation of TGF-β signaling may contribute to the poor differentiation associated with TFEB$^{S142E}$.

To probe how TFEB regulates TGF-β levels, we conducted qRT-PCR and detected no noticeable difference in TGF-β mRNA upon expression of WT/mutant TFEB (Supplementary Fig. 6a), suggesting post-translational regulation of TGF-β upon TFEB expression. Indeed, inhibition of autophagic flux by the lysosomotropic agent chloroquine (CQ), or silencing autophagy essential genes Beclin1 or ATG5, increased TGF-β proteins in A375 cells without affecting their mRNA (Supplementary Fig. 7a–c), indicating a steady-state turnover of TGF-β through the autophagy–lysosomal pathway. TGF-β protein contains two putative LC3-interacting region (xLIR) (Supplementary Fig. 7d), a core consensus motif of (ADEFGLPRSK)(DEGMSTV)(WFY)(DEILQTV)(ADEFHIKLMPSTV)(ILV)[35]. We therefore tested whether LC3 selectively binds TGF-β for lysosomal turnover. Endogenous pro-TGF-β–LC3 interaction was readily detected by co-IP in A375 cells even under basal condition (Supplementary Fig. 7e). TFEB$^{S142A}$ induced autophagy activation and concomitantly promoted pro-TGF-β association with LC3, which was ablated by 3-MA, a PI3KC3 inhibitor that blocks autophagosome biogenesis[36] (Supplementary Fig. 7e). In accord, a significant quantity of TGF-β was present in LC3-labeled autophagosomes and LAMP1$^+$ lysosomes upon TFEB$^{S142A}$ expression, whereas much less was found in TFEB$^{S142E}$ cells (Supplementary Fig. 7f). To determine whether the two xLIR of TGF-β mediate LC3 interaction, we generated the W17A/V20A, F257A/L260A, and W17A/V20A/F257A/L260A mutants of TGF-β (Supplementary Fig. 7g). TGF-β interaction with recombinant LC3 (His-LC3) was

reduced by W17A/V20A or F257A/L260A, and further decreased by W17A/V20A/F257A/L260A (Supplementary Fig. 7g), suggesting that both xLIR motifs are required for efficient LC3 interaction. In vitro experiments using recombinant GST-pro-TGF-β and His-LC3 confirmed their direct interaction, which was disrupted by W17A/V20A/F257A/L260A (Supplementary Fig. 7h). In fact, the suppressed autophagy–lysosome activity in BRAF$^{V600E}$ cells was associated with higher levels of TGF-β protein (not mRNA) as compared with that in MeWo cells (Supplementary Fig. 7i). Disruption of autophagy by Beclin1$^{KD}$ or by Bafilomycin A1 (BafA1) restored TGF-β levels that were suppressed by TFEB$^{S142A}$ (Supplementary Fig. 7j). Corroborating this, TFEB$^{S142A}$-mediated EMT suppression was reversed by BafA1 or by Beclin1$^{KD}$ to a level similar to those observed with TGF-β treatment in the same cells (Fig. 5i). Our results suggest a model where the transcriptional output of TFEB S142 phosphorylation restricts the autophagy–lysosomal-mediated protein turnover of TGF-β, which in turn activates TGF-β signaling and promotes EMT. Thus, through TFEB inhibition, oncogenic BRAF signaling is coupled to TGF-β signaling, promoting melanoma progression and de-differentiation.

**S142 phosphorylation increases melanoma metastatic potential.** We next evaluated the effects of TFEB S142 phosphorylation on the metastasis of BRAF$^{V600E}$ melanoma in an immuno-competent background using the B16-F10 (BRAF$^{WT}$) syngeneic model[37]. We generated B16-F10 melanoma cell lines stably expressing BRAF$^{V600E}$ along with WT or mutant TFEB. TFEB$^{S142A}$ cells showed increased autophagy–lysosome biogenesis, which was suppressed in cells expressing TFEB$^{S142E}$ (Supplementary Fig. 8a, b). Importantly, overexpression of TFEB$^{S142E}$ resulted in a threefold increase in tumor metastasis and favored their lung extravasation after tail vein injection of immuno-competent mice, whereas little/no metastasis was observed upon TFEB$^{S142A}$ expression (Fig. 6a, b). Furthermore, tumor metastasis correlated with suppressed autophagy–lysosomal function, TGF-β signaling activation, and EMT in lung-colonized tumors (Fig. 6a–c). Thus, TFEB S142 phosphorylation confers metastatic properties to BRAF$^{V600E}$ cells.

**Involvement of TFEB in BRAFi-resistant melanoma.** We next determined whether TFEB plays an equivalent role in BRAF$^{V600E}$ melanoma resistant to BRAFi. We examined isogenic A375$^R$ cells that have the BRAF$^{V600E}$ mutation but are BRAFi resistant due to a second NRAS$^{Q61K}$ mutation[38]. Despite their relatively equivalent baseline levels of autophagy–lysosome function, TFEB/ZKSCAN3 status, and EMT-related protein expression to A375 cells, A375$^R$ cells were insensitive to PLX4720, as evidenced by lack of ERK inhibition and JNK2/p38 activation,

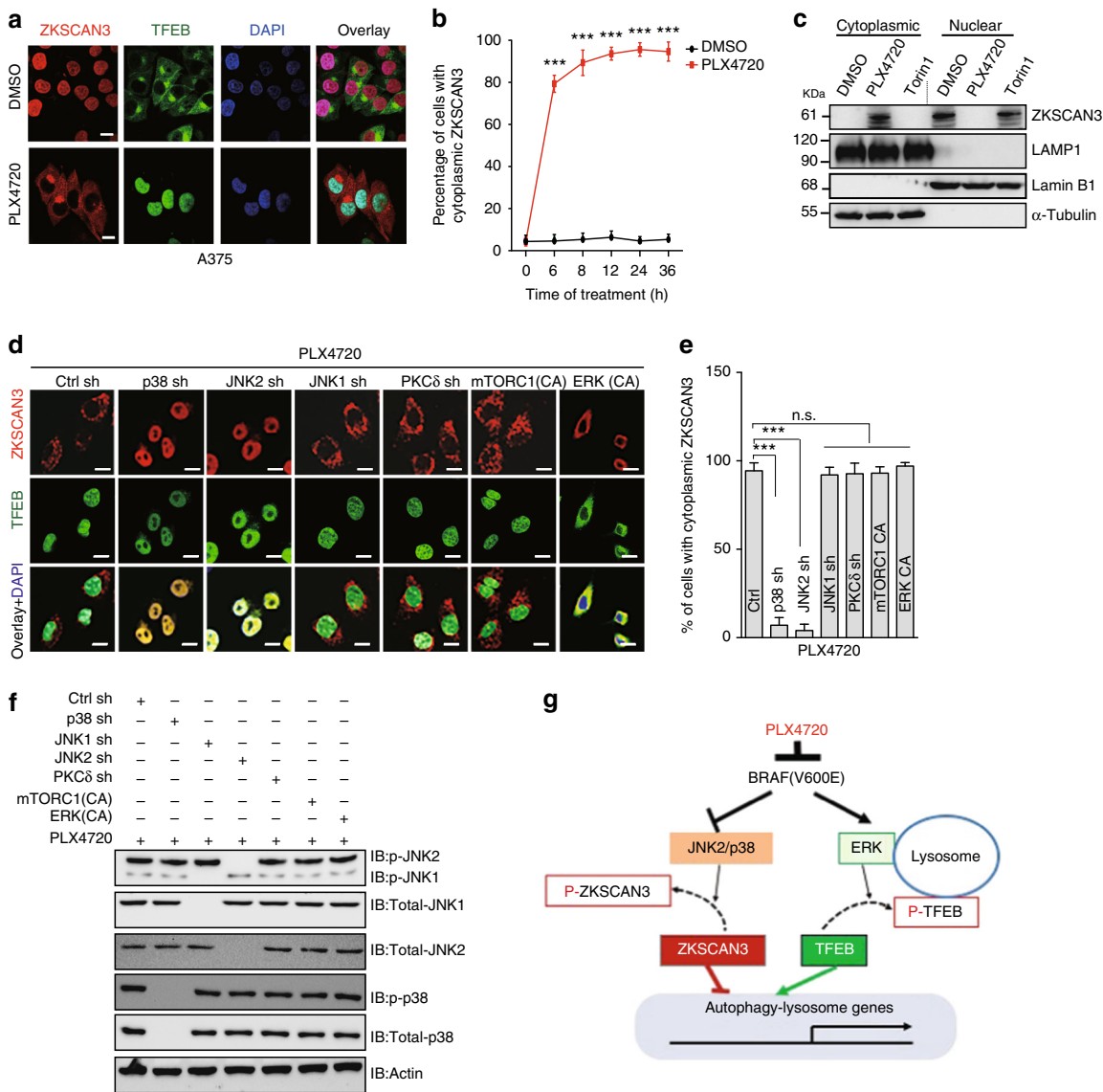

**Fig. 4** PLX4720 induces JNK2/p38 MAPK-dependent cytoplasmic translocation of ZKSCAN3. **a** Representative confocal images of subcellular translocation of endogenous TFEB (green) and ZKSCAN3 (red) in A375 cells treated with PLX4720 (1 μM, 12 h). $n = 3$ independent experiments. **b** Quantification of ZKSCAN3 cytoplasmic translocation in cells shown in (**a**). $n = 150$ cells per time point, pooled from three independent experiments. **c** Immunoblots for ZKSCAN3 in cytoplasmic and nuclear fractions of A375 cells treated with DMSO, PLX4720 (1 μM, 12 h), or Torin1 (1 μM, 3 h). Lamin B1 serves as the control for the nuclear fractions, whereas LAMP1 and Tubulin are the control of the cytoplasmic fractions. Note that PLX4720 induces cytoplasmic enrichment of ZKSCAN3, whereas Torin1 has no effect on its nucleocytoplasmic redistribution. **d** Effect of knockdown of p38, JNK1, JNK2, and PKCδ, or overexpression of the CA form of mTORC1 (E2419K) or ERK (R67S/D321N), on the subcellular translocation of endogenous TFEB (green) and ZKSCAN3 (red) in A375 cells treated with PLX4720 (1 μM, 12 h). Nuclei were stained with DAPI (blue). $n = 3$ independent experiments. **e, f** Quantification of cytoplasmic translocation of ZKSCAN3 (**e**) and immunoblot analyses (**f**) of JNK1/2 and p38 in cells shown in (**d**). $n = 3$ independent experiments. **g** Schematic representation of the dual role of PLX4720 in the transcriptional activation of autophagy and lysosome biogenesis/function: PLX4720 induces TFEB nuclear translocation by inhibition of ERK, whereas also triggering ZKSCAN3 cytoplasmic relocation by activation of JNK2/p38 MAPK, leading to synergistically prolonged activation of the autophagy–lysosomal pathway. See the main text for the details. Scale bars, 10 μm. Data in **c** and **f** are from one experiment that is representative of three independent experiments. For all quantification, data represent the mean ± SD derived from the indicated number of independent experiments. Comparisons were made using Student's $t$-test. \*\*\*$P < 0.001$; n.s. not significant. See Supplementary Fig. 13 for uncropped data of **c**, **f**

TFEB dissociation from 14-3-3, S142 de-phosphorylation, and nuclear translocation, cytoplasmic translocation of ZKSCAN3, autophagy–lysosomal activation, and EMT suppression (Supplementary Fig. 9a b). Nevertheless, as observed in A375 cells, overexpression of TFEB[S142A] decreased colonogenicity and reduced cell proliferation and tumorigenesis of A375[R] cells in xenografts in vivo, whereas TFEB[S142E] had opposite effects (Fig. 7a–d and Supplementary Fig. 9c). Biochemical analyses

confirmed autophagy–lysosomal suppression, increased genomic instability, elevated TGF-β, activated TGF-β signaling, and resultant EMT associated with TFEB[S142E] expression in A375[R] xenografts (Fig. 7e). In contrast, A375[R] xenografts expressing WT, and particularly TFEB[S142A], showed opposite effects (Fig. 7e). These results highlight that the pro-tumorigenic role of TFEB S142 phosphorylation is maintained in BRAFi-refractory melanoma.

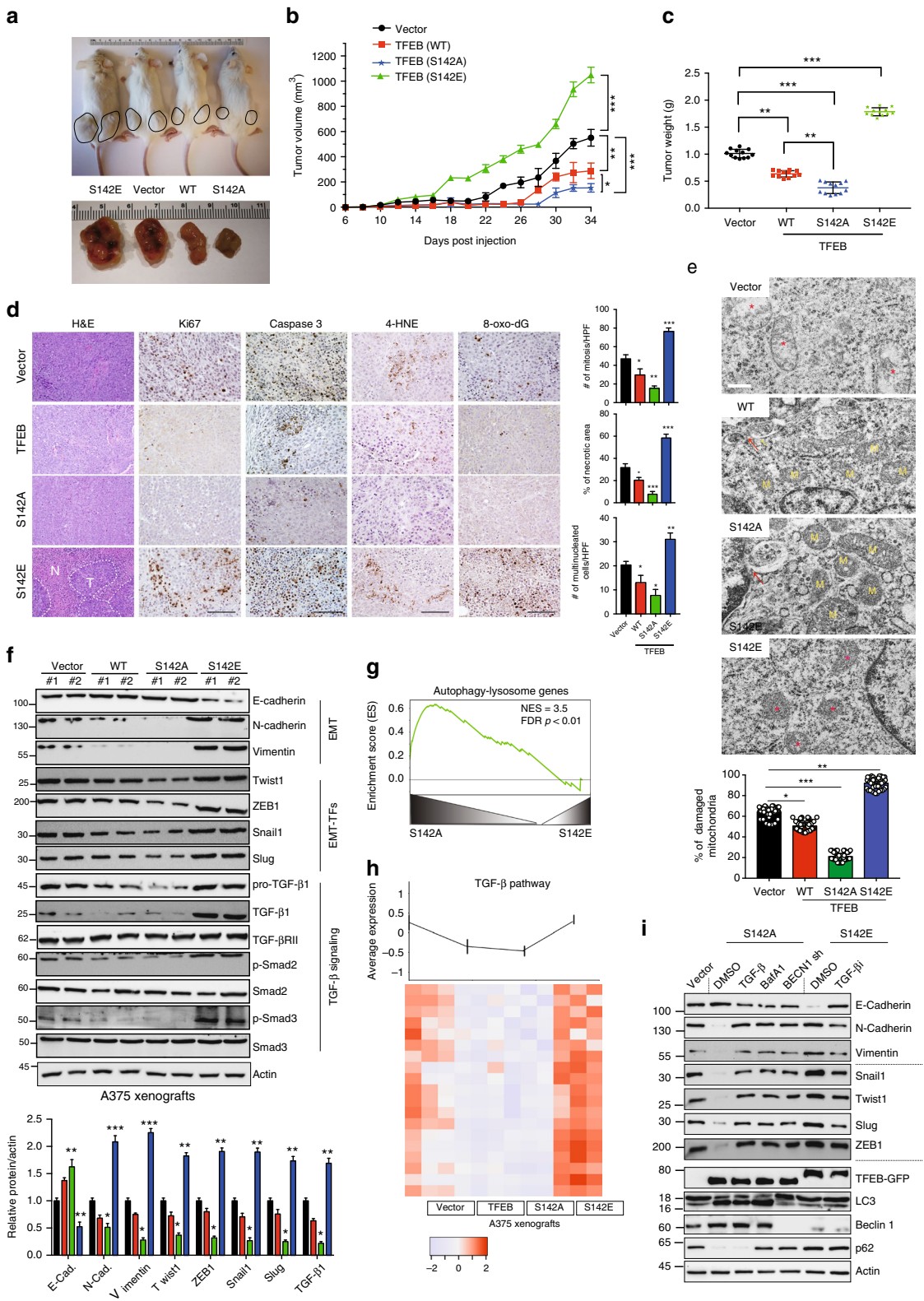

**Autophagy–lysosomal inhibition promotes tumor metastasis**. To investigate whether blocking BRAFi-induced autophagy–lysosomal activation alters melanoma response to BRAF-targeted therapy, we used TFEB[KD] A375 cells reconstituted with WT or mutant TFEB (Fig. 8a). As expected, TFEB KD inhibited PLX4720-induced autophagy–lysosomal activation, which was restored by re-expression of shRNA-resistant WT and

TFEB[S142A], but not TFEB[S142E], whereas ZKSCAN3 expression remained unaffected (Supplementary Fig. 10a). The suppression of PLX4720-induced autophagy–lysosomal activation by TFEB KD or by TFEB[S142E] re-expression caused an increase in both short-term proliferation (Supplementary Fig. 10b) and long-term clonogenicity (Fig. 8a) after PLX4720 treatment as compared with WT- and TFEB[S142A]-complemented cells. This highlights

**Fig. 5** Effect of TFEB S142 phosphorylation on BRAF$^{V600E}$ melanoma progression. **a** NOD/SCID mice bearing A375 xenografts tumors stably expressing vector, WT, TFEB$^{S142A}$ or TFEB$^{S142E}$ isolated on day 34 post-injection. **b**, **c** Tumor volume (**b**) and Tumor weight (**g**) upon autopsy at day 34 of A375 xenografts stably expressing the indicated TFEB. $n = 5$–6 mice per group per time point. **d** H&E and immunohistochemical (IHC) staining of Ki67, active Caspase 3, 4-HNE, and 8-oxo-dG in indicated A375 tumor genotypes. The levels of mitotic figures, necrosis, and multinucleated cells were quantified (right panels; $n = 5$–6 mice per group). N necrotic area, T tumor. Scale bars, 100 μm. **e** EM images of the indicated xenograft tumor genotypes. M mitochondria. Asterisks denote damaged mitochondria. Arrows denote autophagic vacuoles. Damaged mitochondria is quantified (bottom; $n = 50$). Scale bar, 200 nm. **f** IB of EMT-related factors and TGF-β/TGF-βR2/Smad2/3 signaling in indicated A375 xenograft tumor genotypes (two randomly chosen samples per group). Also shown (bottom) is the relative expression of indicated proteins in xenograft tumors. $n = 3$ independent experiments. **g** GSEA plot of A375 tumors showing the significantly changed autophagy–lysosome genes. The relevant complete GSEA data are in Supplementary Data 1a. FDR false discovery rate, NES normalized enrichment score. **h** Heat map depicting the expression of TGF-β target genes in indicated xenograft tumor genotypes. Each column represents a single replicate ($n = 3$). Also refer to Supplementary Data 1d. **i** IB of autophagy and EMT-related factors in A375 cells expressing TFEB$^{S142A}$ or TFEB$^{S142E}$ treated with TGF-β (10 ng/ml, 48 h), BafA1 (100 nM, 6 h), Beclin1 (BECN1)-shRNA, or the TGF-βRI inhibitor SB431542 (TGF-βRIi, 10 μM, 22 h). $n = 3$ independent experiments. Data in **a**, **d**, **e** are from one animal that is representative of 5–6 animals in each group. For all quantification, data represent the mean ± SD derived from the indicated number of independent experiments. Comparisons were made using Student's t-test. *$P < 0.05$; **$P < 0.01$; ***$P < 0.001$. See Supplementary Fig. 13 for uncropped data of **f**, **i**

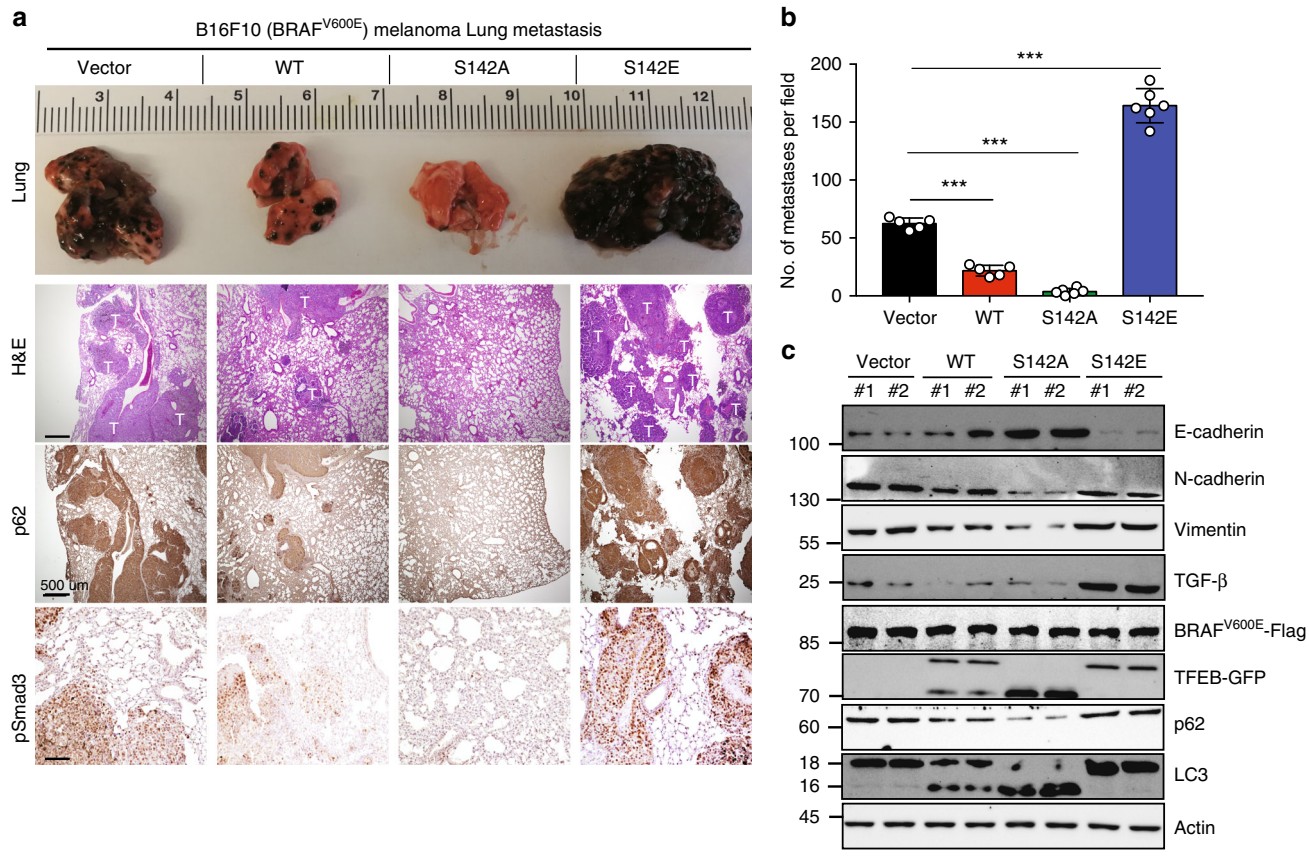

**Fig. 6** TFEB S142 phosphorylation promotes the dissemination of BRAF$^{V600E}$ melanoma cells. **a** Representative gross images (top row), H&E-stained section (second row), and immunohistochemical (IHC) staining of p62 and p-Smad3 in lung metastasis of B16-F10 melanoma cells stably expressing BRAF$^{V600E}$ along with the indicated TFEB proteins. T metastatic melanoma tumors. **b** Quantification of the numbers of lung metastasis formed by B16-F10 melanoma cells as indicated ($n = 5$–6 mice per group; data represent mean ± SD). Comparisons were made (vs. Vector group) using Student's t-test. ***$P < 0.001$. **c** Immunoblot analyses of E-cadherin, N-cadherin, Vimentin, active TGF-β, LC3 conversion, and p62 in indicated lung metastases (two randomly chosen samples per group; similar results were observed in all 10–12 samples per condition). Actin served as a loading control. $n = 3$ independent experiments. See Supplementary Fig. 13 for uncropped data. Scale bars, 500 μm. Data in **c** are from one experiment that is representative of three independent experiments. For all quantification, data represent the mean ± SD derived from the indicated number of independent experiments. Comparisons were made using Student's t-test. ***$P < 0.001$

that TFEB inactivation confers resistance rather than sensitivity to BRAF inhibition in melanoma. Notably, the favorable response of TFEB$^{S142A}$ cells to PLX4720 was abolished when cells were treated with BafA1 or depleted of Beclin1, suggesting that TFEB-induced autophagy–lysosomal activation is required to elicit a response to BRAFi in melanoma (Supplementary Fig. 10c). To validate this, we xenografted NOD/SCID mice with the TFEB-reconstituted clones. When tumors of the same size were palpable, cohorts of mice were treated with PLX4720. Potent growth inhibition and tumor regression was elicited by PLX4720 in TFEB$^{S142A}$ A375 xenografts (Fig. 8b, c) along with increased autophagy–lysosomal activation (Supplementary Fig. 10d–g). In contrast, TFEB depletion or complemented expression of TFEB$^{S142E}$ decreased autophagy–lysosomal activity in xenografts

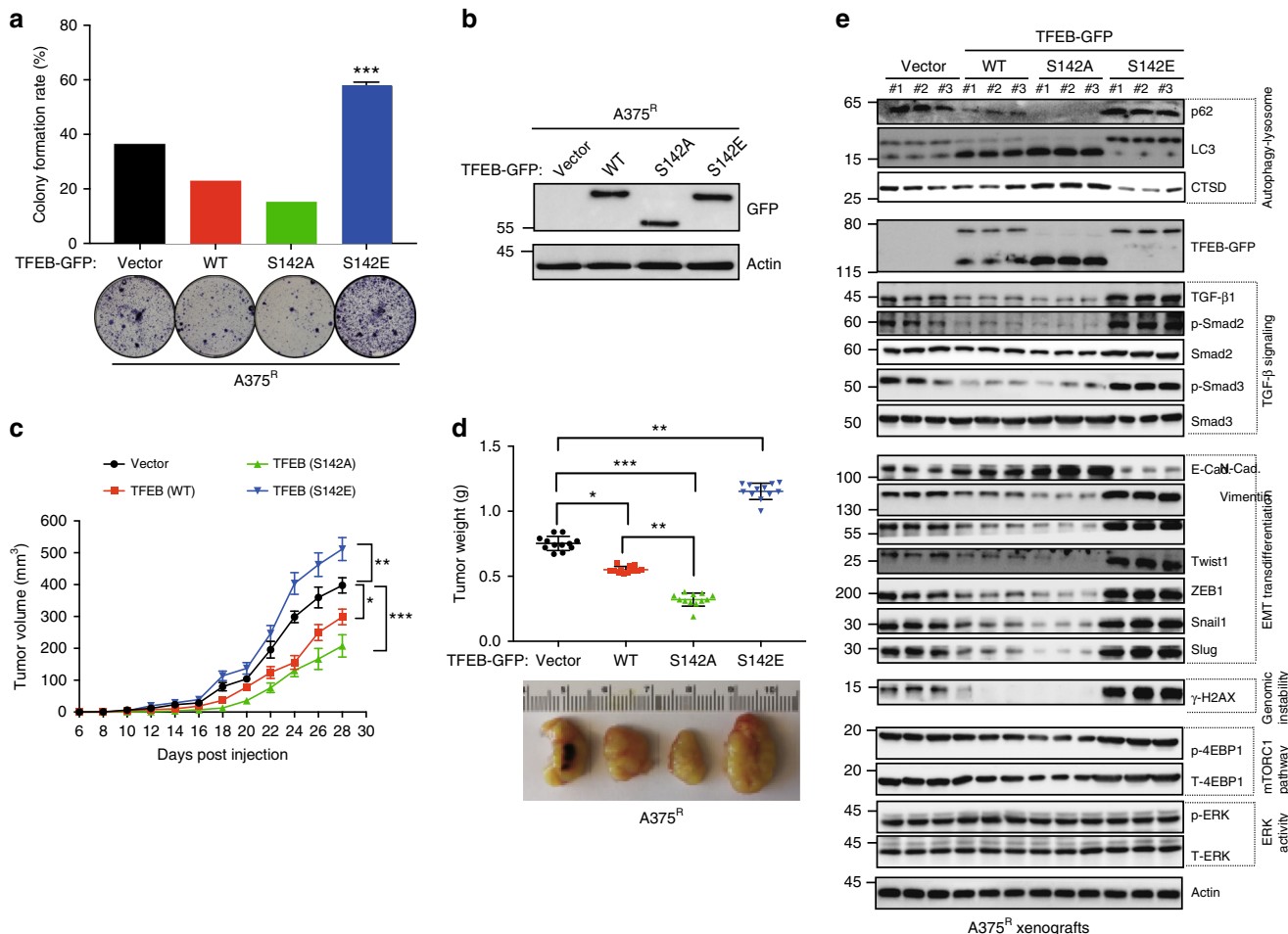

**Fig. 7** Effect of TFEB S142 phosphorylation on PLX4720-resistant BRAF[V600E] melanoma cells. **a** Colony formation assay of PLX4720-resistant A375[R] melanoma cells stably expressing vector, WT TFEB, S142A, or S142E TFEB mutants as indicated. Bars are mean ± SD percentage of colonies for each group after 21 days. $n = 3$ independent experiments. **b** Western blot analysis of TFEB expression in cells shown in (**a**). Actin served as a loading control. Data are from one experiment that is representative of three independent experiments. **c** Tumor volume of xenografts formed after subcutaneous injection of NOD/SCID mice with A375[R] melanoma cells stably expressing the indicated TFEB constructs. Results are the mean volume ± SD for 5–6 mice per group per time point. **d** Tumor weight from experiment shown in (**c**) upon autopsy at day 30. Representative images of tumor size of the indicated A375[R] xenograft tumor genotypes are shown below. **e** Western blot analysis of autophagy (p62 and LC3-I/II), lysosome (CTSD), TGF-β/Smad2/3 signaling, EMT-related [E-cadherin (E-Cad.), N-cadherin (N-Cad.), and Vimentin, Twist1, ZEB1, Snail, and Slug], γ-H2AX, ERK and mTORC1 activation in the indicated xenografts (three randomly chosen samples per group; similar results observed in all 10–12 samples per condition). Actin served as a loading control. See Supplementary Fig. 13 for uncropped data. For all quantification, data represent the mean ± SD derived from indicated number of independent experiments. Comparisons were made using Student's $t$-test. $*P < 0.05$; $**P < 0.01$; $***P < 0.001$

(Supplementary Fig. 10d–g) and led to partial resistance to PLX4720 (Fig. 8b, c). Furthermore, TFEB[S142E] expression resulted in higher incidence of tumor metastasis and colonizations of the liver, lungs, and spleen following PLX4720 treatment, whereas little was found in PLX4720-treated TFEB[S142A] group (Fig. 8d, e and Supplementary Fig. 11b–d). These results indicate that TFEB activation represents a key determinant of the response to monotherapy with BRAFi, and also restricts metastatic dissemination in the course of PLX4720 treatment.

Histological and biochemical analyses of PLX4720-treated A375 xenografts revealed elevated TGF-β and TGF-β signaling (p-Smad3 staining), and promotion of EMT in primary and colonized tumors expressing TFEB[S142E] (Fig. 8d, e; Supplementary Figs. 11a–f). Other pathways that regulate EMT[39] such as Wnt and Notch remained unaffected (Supplementary Fig. 11e, f). Accordingly, PLX4720-treated TFEB[S142E] tumors exhibited a sarcomatoid appearance, as opposed to the polygonal cells in TFEB[S142A] tumors (Supplementary Fig. 11a, b and Fig. 8d), suggesting decreased differentiation, as also

supported by immunoblot analyses (Supplementary Figs. 10d and 11e, f). Notably, no secondary mutations in RAS, MEK, and ERK that could rebound BRAF–MEK–ERK signaling were found in PLX4720-treated TFEB[S142E] tumors, and no concurrent activation of the Akt–mTOR signaling pathway, previously implicated in melanoma resistance[40–42], was detected in PLX4720-treated tumors (Supplementary Figs. 10d and 11e, f). Consistent with the notion that TGF-β activation confers drug resistance in many cancers[43], inhibition of TGF-β signaling by TGF-β receptor inhibitor (TGF-βRi) synergized with BRAFi and re-sensitized TFEB[S142E]-expressing cells to PLX4720 (Supplementary Fig. 10c). On the contrary, addition of recombinant TGF-β compromised the sensitivity of TFEB[S142A] A375 cells to PLX4720 and promoted their clonogenicity, mimicking the effect of autophagy–lysosomal inhibition in the same cells (Supplementary Fig. 10c). Thus, combination of TGF-βRi and BRAFi might be a strategy for treating melanomas with impaired autophagy and/or elevated TGF-β signaling.

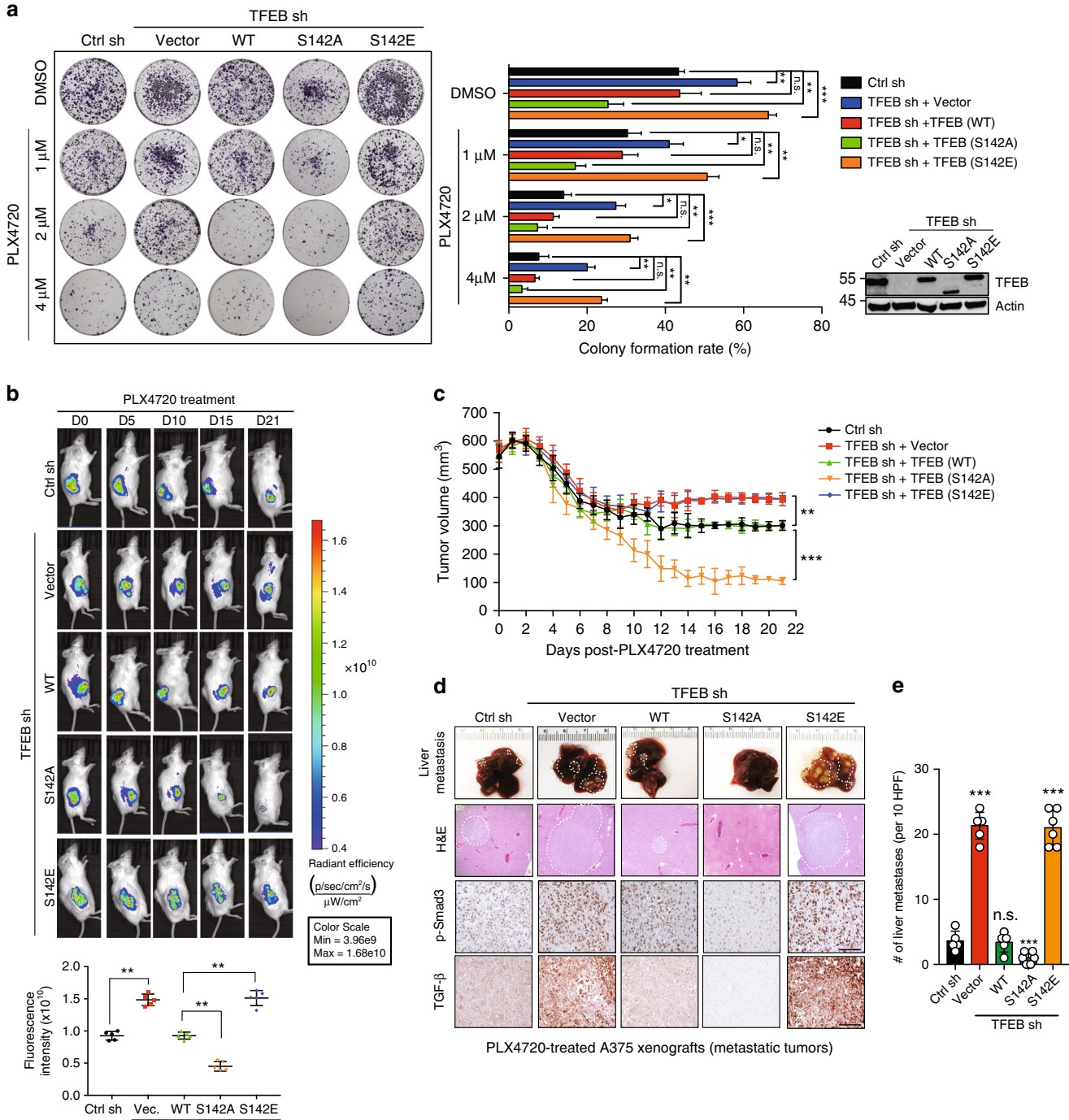

**Fig. 8** TFEB S142 phosphorylation confers BRAFi resistance in melanoma cells. **a** Representative images (left) and quantification (right) of the colonogenic survival of A375 cells stably expressing control shRNA (pGIPZ) or TFEB shRNA complemented with empty vector, WT TFEB, and S142A or S142E TFEB mutants that are shRNA-resistant, after treatment with DMSO or PLX4720 at the indicated concentrations. Endogenous and reconstituted TFEB expression was confirmed by immunoblotting (bottom right). $n = 3$ independent experiments. **b** Bioluminescence images (top) of tumor regression of the indicated A375[R] xenograft tumor genotype in live NOD/SCID mice at the indicated time after inoculation. Radiant efficiency expressed as p/s/cm²/sr/ (µW/cm²) was quantified (bottom). **c** Effect of PLX4720 on tumor response of xenografts formed by the indicated A375 cell lines. After tumor establishment (∼ 500 mm³), mice bearing A375 xenografts were treated with PLX4720 (20 mg/kg, i.p.) daily for 21 days. Values are the mean tumor volume ± SD per time point for 6–7 mice per group. **d**, **e** Representative gross images (**d**) of livers with metastatic nodules (top), H&E-stained sections (second row) and IHC analysis of TGF-β and p-Smad3 of the indicated A375 xenograft genotypes 21 days after PLX4720 treatment (20 mg/kg, i.p., daily). Scale bars, 100 µm. The numbers of metastatic nodules in the liver in (**d**) was quantified in (**e**) ($n = 5$–6 mice per group). For all quantification, data represent the mean ± SD derived from indicated number of independent experiments. Comparisons were made using Student's $t$-test. $*P < 0.05$; $**P < 0.01$; $***P < 0.001$; n.s. not significant

## Discussion

Herein, we show that BRAFi stimulation of autophagy is part of a transcriptional program that coordinates activation of lysosome biogenesis/function mediated by TFEB/ZKSCAN3 (Supplementary Fig. 12). We demonstrate that the BRAF$^{V600E}$–TFEB/ZKSCAN3–autophagy–lysosomal axis represents a key regulatory pathway through which BRAF$^{V600E}$ orchestrates TGF-β signaling and EMT, resulting in tumor progression, metastasis, and resistance to BRAF-targeted therapy in melanoma (Supplementary Fig. 12). Moreover, these data underscore the importance of autophagy–lysosomal function in the control of oncogenic BRAF-associated tumorigenic events.

By examining how autophagy is regulated by BRAFi, we noticed that not only autophagy, but also the lysosomal biogenesis/function, are transcriptionally upregulated in both cell lines and patient specimens, and that this event requires TFEB. In TFEB-deficient melanoma cells, BRAFi could no longer trigger autophagy activation, even though the ER stress machinery remained intact. To understand the functional interaction of TFEB with BRAF$^{V600E}$ signaling, we found that BRAF$^{V600E}$ suppresses TFEB through ERK-mediated TFEB S142 phosphorylation, leading to its cytoplasmic retention and inactivation. Particularly, ERK-mediated S142 phosphorylation, rather than mTORC1-mediated S211 phosphorylation, is a predominant event in BRAF-mutant melanoma. In addition, BRAFi activates JNK2/p38 MAPK, which in turn phosphorylates ZKSCAN3 and allows its cytoplasmic translocation, consequently relieving the repression of TFEB-dependent transcriptome and increasing production of autophagy–lysosomal factors. While our observations do not exclude other reported mechanisms in BRAFi-associated autophagy in melanoma, TFEB/ZKSCAN3 constitutes direct mediators of the autophagy–lysosomal program that affects the responsiveness of melanoma to oncogenic stress.

Upregulation of TFEB has been found in pancreatic tumorigenesis and is required for growth of pancreatic ductal adenocarcinomas[23]. On the other hand, TFEB downregulation is associated with increased colorectal cancer risk and prevention of tumor-associated macrophages activation[44,45]. The discrepancy regarding TFEB in cancer suggests that TFEB may fulfill context-specific roles in distinct cell types. We demonstrate that enhanced TFEB suppression via constitutive expression of a phosphomimetic mutant accelerates BRAF$^{V600E}$ melanoma growth, whereas its non-phosphorylatable derivative behaves in an opposite manner. Importantly, tumors expressing transcriptionally inactive TFEB have biochemical and histopathologic features indicative of a more aggressive and poorly differentiated neoplasm. As BRAF$^{V600E}$ inactivates TFEB in melanoma, it is plausible that endogenous TFEB S142 phosphorylation by the BRAF$^{V600E}$–ERK axis in melanoma similarly contributes to tumor progression.

Although TFEB a broader number of gene expression beyond those in the autophagy–lysosomal pathway, abrogating autophagy–lysosomal function renders BRAF$^{V600E}$-melanoma less responsive to TFEB activation. Furthermore, removing ZKSCAN3 synergizes with TFEB expression in tumor inhibition. Consistently, we observed a correlation between TFEB S142 phosphorylation and autophagy–lysosomal suppression in vitro and in vivo, underscoring a key role of autophagy–lysosomal output of TFEB in melanoma. Indeed, mutational inactivation of TFEB induced increased mitochondrial damage, oxidative stress, tumor necrosis, and genomic instability that have been previously associated with suppressed autophagy in other tumors[46–49]. Thus, a lower rate of autophagy–lysosomal activity within a tumor, as indicated by TFEB inhibition, either at an early stage or during progression, or both, might cause more aggressive disease.

In addition, TFEB S142 phosphorylation induced EMT correlating with metastasis and chemoresistance of xenograft tumors.

To examine how TFEB suppression links BRAF$^{V600E}$ to EMT, we conducted gene expression analysis and found that TGF-β signaling was upregulated upon TFEB inactivation, causing increased EMT. We demonstrated that aberrant activation of TGF-β signaling is due to increased levels of TGF-β that is a selective cargo of autophagy. Inhibition of lysosome or autophagy augments TGF-β levels, even in cells with active TFEB. This study has uncovered a role for TFEB-mediated autophagy–lysosomal activation in suppressing TGF-β signaling, and showed that TFEB inactivation enhances TGF-β secretion/signaling. Thus, BRAF$^{V600E}$-dependent TFEB phosphorylation and TGF-β activation enhances tumor aggressiveness (Supplementary Fig. 12), as observed in BRAF-mutant melanoma patients[50,51].

The belief that autophagy inhibition promotes tumor inhibition and chemosensitivity is largely based on studies of antitumor effects of the lysosomotropic agent CQ or its derivatives in combination chemotherapy[6,52,53]. However, it should be noted that CQ, which virtually affects most acidic compartments in cells, is not a selective autophagy inhibitor, and that CQ exhibits antitumor activities beyond autophagy and even lysosomal functions[1,54]. It is also worth noting that CQ treatment alone is sufficient to induce TFEB nuclear translocation and activation of lysosomal and autophagy genes[55,56], which may contribute to tumor regression in combination therapy. Thus, the mechanistic underpinnings of CQ-based trials await further investigation. Nevertheless, our data demonstrate that blockade of BRAFi-induced autophagy–lysosomal activation adversely (rather than favorably) affects oncological outcomes by aberrant activation of TGF-β pathway. Moreover, inhibition of TGF-β signaling reverses resistance of TFEB$^{S142E}$ cells to BRAFi. Conversely, the favorable response of TFEB$^{S142A}$ cells to BRAFi is abolished by the absence of autophagy–lysosomal machinery or by TGF-β supplementation. Consistent with our findings, a synergistic interaction between inhibition of BRAF and TGF-β has been observed in BRAF-mutant cancers[57]. Altogether, we have identified an unexpected activity of TFEB-mediated autophagy–lysosomal function in regulating TGF-β signaling, which when compromised, promotes tumor progression and drug resistance in BRAF-mutant melanoma. In light of this, targeting autophagy for cancer therapy may need to be re-visited.

## Methods

**Cell culture and transfection.** All cell lines were purchased from the American Type Culture Collection (ATCC) and maintained at 37 °C with 5% CO₂. HEK293T (ATCC®CRL-3216), A375 (ATCC®CRL-1619), G361 (ATCC®CRL-1424), MeWo (ATCC®HTB-65), SK-MEL-5 (ATCC®HTB-70), SK-MEL-2 (ATCC®HTB-68), B16 (ATCC®CRL-6475), and HT29 (ATCC®HTB-38) cells were cultured in Dulbecco's modified Eagle's medium (DMEM) supplemented with 10% fetal bovine serum (Gibco, 10437028), 2 mM L-glutamine, and 1% penicillin–streptomycin (Invitrogen, 15140122). Transfections were performed using Calcium Phosphate Transfection Kits (Clontech, 631312) or PolyFect Reagent (Qiagen, 301107), following the manufacturer's instructions. None of the cell lines used in this study was found in the database of commonly misidentified cell lines that is maintained by ICLAC and NCBI Biosample. All cell lines were tested and confirmed to be free of mycoplasma.

**Plasmids.** The full-length complementary DNA (cDNA) clones of human TFEB (plasmid #99955), BRAF$^{V600E}$ (plasmid #15269), mTOR (E2419K) (plasmid #19994), ERK (R67S/D321N) (plasmid #53203), and RagB (Q99L) (plasmid #19315), were purchased from Addgene (USA). The Flag-, V5-, or GFP-tagged WT TFEB, ZKSCAN3, pro-TGF-β, and their mutant derivatives were constructed by cloning the cDNA of the full-length or point mutants into pcDNA5/FRT/TO, pcDNA3.1 (for pro-TGF-β), or into pCDH-CMV-MCS-EF1-Puro vector to generate lentiviral transfer constructs. Flag-tagged BRAF$^{V600E}$ was cloned into pCDH-CMV-MCS-EF1-puro backbone (System Biosciences; SBI) to generate lentiviral transfer constructs. shRNA-resistant TFEB mutants were generated using site-directed mutagenesis (Q5 0552S, New England Biolab). For expression of GST fusion in *Escherichia coli* BL21 cells, the full-length TFEB, pro-TGFβ, and its point mutants were cloned into the pGEX-4T-1 (Pharmacia Amersham) vector or the pET32a (EMD Biosciences) vector. The plasmids of ZKSCAN3-V5 was provided by Dr. Pinghui Feng, University of Southern California, USA. All constructs were

confirmed by sequencing using an ABI PRISM 377 automatic DNA sequencer (Applied Biosystems, CA).

**Antibodies and other reagents.** The following antibodies were used in this study at the indicated dilution for western blot (WB) analysis, immunoprecipitation (IP), immunohistochemistry (IHC), and immunofluorescence (IF): TFEB (PA5-34360, ThermoFisher; 1:100 for IP, 1:200 for IF, 1:1000 for WB), TFE3 (ab179804, Abcam; 1:200 for IF, 1:1000 for WB), TFEB-pS142 (ABE1971, EMD-Millipore; 1:2000 for WB), MITF-M (MA5-14146, ThermoFisher; 1:200 for IF, 1:1000 for WB), LC3 (CAC-CTB-LC3-2-IC, Cosmo Bio USA; 1:100 for IF; #2775S, Cell Signaling; 1:1000 for WB), p62 (#5114S, Cell Signaling, 1:1000 for WB, 1:500 for IHC), Cathepsin D (#2284, Cell Signaling, 1:1000 for WB), LAMP1 (14968, BD, 1:200 for IF, 1:1000 for WB), GRP78 (ab121390, Abcam, 1:1000 for WB), PERK (#5683T, Cell Signaling, 1:1000 for WB), Lamin B1 (13435S, Cell Signaling, 1:1000 for WB), p-ERK (T202/Y204) (#4370, Cell Signaling, 1:1000 for WB), ERK (#4695, Cell Signaling, 1:200 for IF, 1:1000 for WB), p-p70S6K(T389) (#9206S, Cell Signaling, 1:1000 for WB), p70S6K (#9202S, Cell Signaling, 1:1000 for WB), p-4E-BP1(T37/46) (#9451T, Cell Signaling, 1:1000 for WB), 4E-BP1 (#9644S, Cell Signaling, 1:1000 for WB), p-GSK3α(S21) (#9327, Cell Signaling, 1:1000 for WB), GSK3α (#4337, Cell Signaling, 1:1000 for WB), p-GSK3β(S9) (#9327, Cell Signaling, 1:1000 for WB), GSK3β (#9315, Cell Signaling, 1:1000 for WB), p-Akt (T308) (#13038, Cell Signaling, 1:1000 for WB), p-Akt (S473) (#4060, Cell Signaling, 1:1000 for WB), Akt (#4685, Cell Signaling, 1:1000 for WB), p-JNK1/2 (#9255S, Cell Signaling, 1:1000 for WB), JNK1/2 (#9258, Cell Signaling, 1:1000 for WB), p-p38 (#9215S, Cell Signaling, 1:1000 for WB), p38 (#9212S, Cell Signaling, 1:1000 for WB), RSK (#9355, Cell Signaling, 1:1000 for WB), MCOLN1 (SAB1407780, Sigma-Aldrich, 1:1000 for WB), Importin 8 (NBP2-24751, Novus Biologicals, 1:1000 for WB), CRM1 (#46249, Cell Signaling, 1:1000 for WB), DEPDC5 (GTX33570, GeneTex, 1:500 for WB), 14-3-3(pan) (#8312, Cell Signaling, 1:200 for IF, 1:1000 for WB), p-14-3-3 motif (#9601, Cell Signaling, 1:1000 for WB), ZKSCAN3 (NBP1-31566, Novus Biologicals, 1:200 for IF, 1:1000 for WB), mTOR (#2983, Cell Signaling, 1:1000 for WB, 1:200 for IF), PKCδ (#9616, Cell Signaling, 1:1000 for WB), Ki67 (NB110-89719, Novus Biological, 1:500 for IHC), cleaved caspase 3 (#9661T, Cell Signaling, 1:100 for IHC, 1:1000 for WB), PARP (#9532, Cell Signaling, 1:1000 for WB), 4-HNE (ab46545, Abcam, 1:600 for IHC), 8-oxo-dG (bs-1278R, Bioss, 1:100 for IHC), E-cadherin (20874-1-AP, Proteintech, 1:1000 for WB), N-cadherin (GTX-127345, GenTex, 1:1000 for WB), Vimentin (#5741T, Cell Signaling, 1:1000 for WB), Twist1 (#46702S, Cell Signaling, 1:1000 for WB), ZEB1 (#3396T, Cell Signaling, 1:1000 for WB, Snail1 (#3879T, Cell Signaling, 1:1000 for WB), Slug (#9585T, Cell Signaling, 1:1000 for WB), p-Smad2 (#18338, Cell Signaling, 1:1000 for WB), Smad2 (#5339, Cell Signaling, 1:1000 for WB), p-Smad3 (#9520, Cell Signaling, 1:150 for IHC, 1:1000 for WB), Smad3 (#9523, Cell Signaling, 1:1000 for WB), pro-TGF-β (#3711, Cell Signaling, 1:1000 for WB), TGF-β1 (21898-1-AP, Proteintech, 1:1000 for IHC, 1:1000 for WB, 1:50 for IF), TGF-βRII (#2518, Cell Signaling, 1:1000 for WB), Beclin1 (#3495, Cell Signaling, 1:1000 for WB), ATG5 (#12994, Cell Signaling, 1:1000 for WB), Actin (sc-47778, Santa Cruz; 1:1000 for WB), GFP (GTX-113617, GenTex, 1:100 for IP, 1:200 for IF, 1:1000 for WB), Flag (F3165, Sigma-Aldrich; 1:100 for IP, 1:200 for IF, 1:1000 for WB), Flag (F2555, Sigma-Aldrich; 1:100 for IP, 1:200 for IF, 1:1000 for WB), HA (NB600-363, Novus; 1:1000 for WB), V5 (NB600-381, Novus; 1:1000 for WB). Horseradish peroxidase (HRP)-labeled or fluorescently labeled secondary antibody conjugates were purchased from Molecular Probes (Invitrogen). Purified rabbit IgG was purchased from Pierce. Purified ERK(CA) proteins were purchased from Millipore Sigma (#14-550). LysoTracker Red DND-99 (0.3 μM; L7528, ThermoFisher) was used to stain the lysosome compartments. Inhibitors or activators used in this study include PLX4720 (ab141362, Abcam; 1 μM or as indicated), Torin1 (Selleckchem S2827, 1 μM, 3 h), BafA1 (Sigma B1793, 100 nM, 6 h), CQ (C6628, Sigma, 20 μM, 24 h), ERK inhibitor FR180204 (Selleckchem S7524, 10 μM, 24 h), RSK inhibitor (SL0101; 50 μM, 24 h), CRM1 inhibitor LMB (L2913, Sigma-Aldrich, 20 nM, 2 h), PI3KC3 inhibitor 3-MA (M9281, Sigma, 100 nM, 24 h), TGF-βRI inhibitor SB431542 (Sigma-Aldrich S4317; 10 μM, 22 h), TGF-β1 (Cell Signaling #8915; 10 ng/ml, 48 h). Unless otherwise stated, all chemicals were purchased from Sigma-Aldrich.

**IF and confocal microscopy.** Cells plated on coverslips were fixed with 4% paraformaldehyde (20 min at room temperature (RT)). After fixation, cells were permeabilized with 0.2% Triton X-100 for 8 min and blocked with 10% goat serum (Gibco-BRL) for 1 h. Primary antibody staining was carried out using antiserum or purified antibody in 1% goat serum for 1–2 h at RT or overnight at 4 °C. Cells were then extensively washed with phosphate-buffered saline (PBS) and incubated with diluted Alexa 488-, Alexa 594-, and/or Alexa 633-conjugated secondary antibodies in 1% goat serum for 1 h, followed by DAPI (4', 6'-diamidino-2-phenylindole) staining. Cells were mounted using Vectashield (Vector Laboratories, Inc.). Confocal images were acquired using a Nikon Eclipse C1 laser-scanning microscope (Nikon, PA), fitted with a 60× Nikon objective (PL APO, 1.4 NA), and Nikon imaging software. Images were collected at 512 × 512 pixel resolution. The stained cells were optically sectioned in the z axis. For multichannel imaging, fluorescent staining was imaged sequentially in line-interlace modes to eliminate crosstalk between the channels. The step size in the z axis varied from 0.2 to 0.5 mm to obtain 16 slices/imaged file.

For image quantification, approximately 200 cells, randomly chosen from 10 high-power fields (HPFs) and pooled from three independent experiments, were evaluated for the distribution pattern of the indicated molecules. The Pearson correlation coefficient was calculated using the built-in colocalization analysis module of the NIS-Elements AR software. All experiments were independently repeated several times. The investigators conducted blind counting for each quantification-related study.

**Histopathology and IHC.** Tissue sections were fixed in 10% buffered formalin and embedded in paraffin. Tissue sections were routinely stained with hematoxylin and eosin. For IHC staining, tissue slides were deparaffinized in xylene and rehydrated in alcohol. Endogenous peroxidase was blocked with 3% hydrogen peroxide. Antigen retrieval was achieved using a hot water bath and 10 mM citric sodium buffer (pH 6.0). Sections were then incubated overnight at 4 °C with the indicated primary antibody. Antibody binding was detected with EnVision™ Dual Link System-HRP DAB kit (K4010, Dako). Sections were then counterstained with hematoxylin. For negative controls, the primary antibody was excluded. The mitotic index was quantified by viewing and photographing 10 random HPFs for each tissue section on a Keyence All-In-One Fluorescence Microscope, using a 40× or 20× objective. For evaluation and scoring of immunohistochemical data, we randomly selected 10 fields within the tumor area under high-power magnification (×40) for evaluation. The investigators conducted blind counting for all quantification.

**Conventional EM.** A375 cells were fixed overnight at 4 °C in 1/2 strength Karnovsky's (2% paraformaldehyde and 2.5% glutaraldehyde in 0.2 M sodium cacodylate buffer, pH 7.4). Cells were rinsed in 0.1 M sodium cacodylate buffer and pelleted. Cell pellets were treated with 2% osmium tetroxide in 0.2 M cacodylate buffer for 2 h at 4 °C, and rinsed in 0.1 M cacodylate buffer. Samples were then blocked, and stained with 1% uranyl acetate overnight at 4 °C. Pellet was then rinsed with 0.1 M sodium acetate. Samples were dehydrated through a graded series of ethanol, and then infiltrated with Epon resin overnight at room temperature. They were then embedded in resin overnight at 60 °C. Thin sections were cut on a Sorvall MT 6000 ultramicrotome and collected onto copper grids. Sections were examined on a JEOL 2100 transmission EM. Images were recorded on film at ×5000 magnification. The entire population of mitochondria in 20 images was examined to count the number of abnormal mitochondria. The percentage of abnormal mitochondria was determined by dividing the number of abnormal mitochondria by the total number of mitochondria per image.

**Immunoblotting and IP.** For IP, cells were washed with ice-cold PBS, lysed in 2% Triton X-100 lysis buffer (20 mM Tris at pH 7.5, 150 mM NaCl, 1 mM EDTA and 2% Triton X-100) supplemented with a phosphatase inhibitor mix (Pierce) and a complete protease inhibitor cocktail (Roche). After sonication (Misonnix ultrasonic S-4000, amplitude 15%, process time 10 s, push-on time 5 s, and push-off time 1 s), cell lysates were rotated at 4 °C for at least 30 min. The soluble fraction was isolated by centrifugation at 21,130 × g for 10 min at 4 °C, and subjected to pre-clearing with protein A/G agarose beads for 1 h at 4 °C. Whole-cell lysates (WCLs) were used for IP with the indicated antibodies. Generally, 1–4 μg commercial antibody was added to 1 ml WCL, which was then incubated at 4 °C for 8–12 h. After addition of protein A/G agarose beads, incubation was continued for another 2 h. Immunoprecipitates were extensively washed with IP wash buffer (10 mM Tris at pH 7.5, 150 mM NaCl, 1 mM EDTA, 0.2% Triton X-100) supplemented with 1× phosphatase inhibitor mix (Pierce) and 1× protease inhibitor mix (Roche), and then eluted with SDS–PAGE loading buffer by boiling for 5 min. For immunoblotting, polypeptides were resolved by sodium dodecyl sulfate–polyacrylamide gel electrophoresis (SDS–PAGE) and transferred to a polyvinylidene difluoride (PVDF) membrane (Bio-Rad). Membranes were blocked with 5% non-fat milk or bovine serum albumin (BSA), and probed with the indicated antibodies. HRP-conjugated goat secondary antibodies were used (1:3000, Invitrogen). Immunodetection was achieved with Hyglo chemiluminescence reagent (Denville Scientific), and detected by a Bio-Rad ChemiDoc machine.

**Autophagy analyses.** Autophagy was measured by light microscopy quantitation of numbers of GFP-LC3 puncta per cell in cells transfected with GFP-LC3 or by WB analysis based on the ratio of LC3-II/LC3-I and amount of p62/actin[58,59]. Starvation was induced by treating cells with HBSS (Corning 21021149) for 6 h. All GFP-LC3 puncta quantitation was performed by observers blinded to experimental conditions.

**β-NAG assay.** NAG assays were performed using a kit from Sigma-Aldrich (CS0780) following the manufacturer's instructions. Briefly, A375 cells treated with PLX4720 (1 μM, 12 h) were lysed in 1X RIPA lysis buffer (150 mM sodium chloride, 1% Triton X-100, 1% sodium deoxycholate, 0.1% SDS, 50 mM Tris-HCl, pH 7.5, and 2 mM EDTA). Cell lysates (10 μg) from each sample were normalized to equal volume and measured in triplicate for NAG activity following the protocol provided by the supplier.

**GST fusion protein purification**. For TFEB and pro-TGF-β protein purification, the full-length cDNA fragment of TFEB (WT, S142A, S211A) or pro-TGF-β (WT, W17A/V20A, F257A/L260A, W17A/V20A) was cloned into the pGEX-4T-1 vector or pET32a vector. The resulting plasmids were used to transform the BL21 (DE3) bacteria strain and the GST–TFEB protein was induced by 0.5 mM IPTG. Cells were collected by centrifugation and resuspended in PBS buffer with protease inhibitors (Roche), followed by sonication for 10 s at amplitude 15%. Cell debris were spun down and the clarified supernatant was loaded onto a Glutathione-Agarose (Sigma G4510) column under gravity flow. After four PBST (PBS containing 1% Triton X-100) washes, proteins were eluted with elution buffer (10 mM reduced glutathione in 50 mM Tris-HCl, pH 9.0) and analyzed by electrophoresis on 8% SDS-polyacrylamide gels followed by Coomassie blue staining.

**Subcellular fractionation**. Cells were lysed in NP40 buffer (25 mM Tris at pH 7.5, 300 mM NaCl, 1 mM EDTA and 2% NP40) for 15 min on ice. The lysates were centrifuged at $1000 \times g$ for 5 min. The supernatant contains the cytosolic and membrane fractions. The pellets were resuspended in NP40 lysis buffer and sonicated three times for 5 s each at 20% power to release nuclear proteins.

**In vitro kinase assay**. Purified recombinant GST-TFEB WT or mutants immobilized on Glutathione-Agarose beads (Sigma G4510) were incubated with purified ERK (2 μg) for 2 h at 30 °C in kinase buffer (25 mM Tris-HCl (pH 7.5), 5 mM beta-glycerophosphate, 2 mM dithiothreitol (DTT), 0.1 mM Na₃VO₄, 10 mM MgCl₂ (Cell Signalling Technology #9802) containing 0.5 μCi of [γ-³²P]ATP (Perkin Elmer). The beads were spun down, washed three times with PBS, and eluted with SDS sample buffer (Sigma S3401) and resolved by SDS–PAGE. The gels were dried and exposed to phosphor-imager (Fujifilm FLA-5000) screens for autoradiography, followed by Coomassie blue staining to visualize the proteins.

**Lentiviral gene KD by shRNA**. All shRNAs were purchased from Open Biosystem. Lentiviral-compatible shRNAs against TFEB (sh1: V3LHS_332989, sense: TGTTGG TCATCTCCAGGCG; sh2: V3LHS_332992, sense: TCGCTAGGCAGCTCCTGCT), TFE3 (sh1: V2LHS_197759, sense: ATTGTAACTGGACTCCAGG; sh2: V3LHS_357540, sense: ATGACATCATCAATCTCCT), MITF (sh1: V2LHS_257541, sense: TAACCTATTAATACTACAC; sh2: V2LHS_259964, sense: ATTCTTTCTAGAAA GCCTG), GRP78 (sh1: V3LHS_380915, sense: CTCTGTGTCCACAGAGCCG; sh2: V3LHS_380916, sense: TAGCAATGCCAATCTTCCT), PERK (sh1: V2LHS_68173, sense: TCTTACATCAGTTAAGGTC; sh2: V2LHS_68177, sense: TATACCGAAGT TCAAAGTG), ERK (sh1: V2LHS_217986, sense: ACTTCAATCCTCTTGTGTG; sh2: V2LHS_47250, sense: TAAGTCATTACATAATGCC), RSK1 (sh1: V2LHS_241402, sense: TCTCTTCTGAAGGATCCCG; sh2: V2LHS_47379, sense: ACCTCT ACCAAGATATCAC), JNK1 (sh1: V2LMM_49133, sense: ATTACTAGGCTTTA AGTCC; sh2: V3LMM_420425, sense: TTTGGATAACAAATCTCTT), JNK2 (sh1: V2LHS_170511, sense: TAATACCACAAAGCATCTG; sh2: V2LHS_170513, sense: AGTTTCTTCATGAACTCTG), p38 (sh1: V2LHS_113215, sense: TTCATATGTT TAAGTAACC; sh2: V2LHS_113218, sense: TTCACAGCTAGATTACTAG), MCOLIN1 (sh1: V2LHS_249668, sense: TATTGATGAGGCTCTGGAG; sh2: V3LHS_338978, sense: AGATGACAGCCACGCAGCA), IPO8 (sh1: V2LHS_198074, sense: ATAGTATACCTAGAAGCTG; sh2: V2LHS_5805, sense: TAAAGACTGA AGAGCAAGG), PKCδ (V2LMM_62352, sense: TTCTCATTCAGGAACTCTG), Raptor (sh1: V2LMM_63671, sense: ATCAGAAACATCTGGATAG; sh2: V2LMM_73881 sense: TTCTAAACAAACTTGCCAC), DEPDC5 (sh1: V2LHS_261672, sense: TTCTGCATGATGTCAATGG; sh2: V2LHS_79321, sense: TCTGCAAGCCTTTCA TGGC), CRM1 (sh1: V2LHS_172053 sense: TTGACAGAGACTTTCGCTG; sh2: V2LHS_172054 sense: AAGATAAACCAATGTTTCC), ATG5 (V2LHS_248503, sense: TATCTCATCCTGATATAGC), ZKSCAN3 (V3LHS_351690, sense: AGTCTG TTTTTCATCACCC), Beclin1 (V3LHS_349509, sense: TTTCTGCCACTATCT TGCG).

For lentivirus production, HEK293T cells were transfected with the transfer vector (e.g., pCDH-CMV-MCS-EF1-Puro or pGIPZ), pCMV-dR8.91 packaging plasmid, and pCMV-VSV-G envelope plasmid in a 5:1:4 ratio using the Calcium Phosphate Transfection Kit (Clontech). The medium was replaced 12 h later. Viral particles were collected 48 h post-transfection, filtered with 0.45 μm sterile filter, and concentrated overnight by Lenti-X concentrator (631312, Takara) at a ratio of 3:1, followed by centrifugation at 4 °C (28,800 × g, 2 h, ThermoFisher Sorva RC 6+). Viral particles were resuspended in fresh medium with 8 μM/mL polybrene, and were plated with target cells for 24 h. Lentiviral-transduced cells were selected in 2 μg/mL puromycin for 7 days with the medium changed daily.

**RNA extraction, cDNA synthesis, and qPCR analysis**. Total RNA was isolated with RNeasy Mini Kit (Qiagen 74104), and 1 μg of total RNA was used for cDNA synthesis using iScript™ cDNA Synthesis Kit (Bio-Rad). Quantitative real-time PCRs were carried out using the primers listed in Supplementary Data 2 and iQ SYBR Green Master Mix (Bio-Rad). Samples were obtained and analyzed on the CFX96 Touch Real-Time PCR Detection System. The gene expression levels were normalized to actin. Primers used for qPCR are included in Supplementary Data 2.

**Clonogenic cell survival assay**. The log-phase cells were plated in six-well plates overnight allowing cells to attach to the plates. After PLX4720 treatment (24-h

exposure), cells were trypsinized, counted, and re-plated at appropriate dilutions for colony formation. After 10–14 days of incubation, colonies were fixed with methanol:acetic acid (3:1), stained with crystal violet, and counted. Plating efficiency (PE) was determined for each individual cell line as described[60], and the surviving fraction (SF) was calculated based on the number of colonies that arose after treatment, expressed in terms of PE. Each experiment was repeated three times.

**Animal experiments**. NOD/SCID mice (JAX Stock Number 001303) and C57Bl/6J mice (JAX Stock Number 000664) were purchased from Jackson Laboratory (Bar Harbor). All animal studies were performed in compliance with the University of Southern California Institutional Animal Care and Use Committee (IACUC) guidelines.

For subcutaneous xenografts, A375 cells stably expressing WT TFEB or TFEB mutants ($5 \times 10^6$) were injected subcutaneously into the lower flank of 6-week-old female NOD/SCID mice. Mice were monitored tri-weekly for the development of tumors by measurements of tumor weight, tumor length (L), and width (W); tumor volume was calculated according to the formula (length × width²)/2 as described[61]. After a 3-week observation period (post-inoculation), mice were sacrificed and tumors were dissected; half of each tumor was frozen in liquid nitrogen and half was fixed in 4% paraformaldehyde for subsequent histological examination.

For lung metastasis, B16-F10 melanoma cells ($5 \times 10^5$) stably expressing BRAF^V600E along with WT or mutant TFEB proteins were injected via the lateral tail vein of C57Bl/6J (6-week-old, age and gender matched) mice using a 27-gauge needle. Mice were sacrificed 14 days after injection and tissues were isolated and fixed in 10% neutral-buffered formalin. Surface metastatic foci in lung lobes were counted under a dissecting microscope. For quantitation of metastatic nodule size, photos of random fields were obtained and then the sizes of at least 20 nodules were determined using NIH Image software and averaged.

To measure the response of melanoma expressing WT or mutant TFEB to PLX4720, NOD/SCID mice (5–6 mice per group) were injected subcutaneously with $10^7$ tumor cells in 200 μl of PBS as described above. Once tumors grew to a palpable size (~ 500 mm³), mice were randomized and treated daily with 20 mg/kg (body weight) PLX4720 via intraperitoneal injection. Tumor size was measured with a caliper daily until 21 days after treatment initiation. Mice spleens, livers, lungs, and lymph nodes were removed and examined for tumor metastases. The tissues were formalin-fixed and paraffin-embedded for histological analysis. Alternatively, GFP-labeled A375 cells could be tracked using bioluminescence imaging during the course of treatment. Briefly, mice were placed in the induction chamber with 2% isoflurane in oxygen. GFP activity was localized and quantified using an IVIS III image system. Images were taken with an excitation wavelength of 465 nm and emission wavelength ranging from 500 to 540 nm. Image processing and analysis, including flat fielding, adaptive background subtraction and spectral unmixing were performed with Living Image® 3.0 software.

**High-throughput RNA sequencing (RNA-seq)**. For RNA isolation and library preparation, RNA was prepared using the Qiagen RNeasy Kit (Qiagen, Valencia, CA) according to the manufacturer's instructions. Strand-specific library perpetration was carried out using a KAPA Stranded mRNA-Seq Kit, with KAPA mRNA Capture Beads (KAPA Biosystem Wilmington, MA). Sequencing libraries were validated using the Agilent Tapestation 4200 (Agilent Technologies, CA, USA), and quantified by using Qubit 2.0 Fluorometer (Invitrogen, Carlsbad, CA), as well as by qPCR (Applied Biosystems, Carlsbad, CA, USA). The sequencing libraries were multiplexed and clustered on one lane of a Flow Cell. After clustering, the flow cell was loaded on the Illumina HiSeq instrument according to manufacturer's instructions. The samples were sequenced using a 2 × 150 Paired End (PE) configuration. Image analysis and base-calling were conducted by the HiSeq Control Software (HCS). Raw sequence data (.bcl files) generated from Illumina HiSeq were converted into Fastq files and de-multiplexed using Illumina's bcl2fastq 2.17 software. One mis-match was allowed for index sequence identification. RNA-seq was performed by GENEWIZ Plainfield (NJ, USA).

For bioinformatic analyses of RNA-seq, the quality of Fastq files was checked with FastQC (Baraham Bioinformatics group, http://www.bioinformatics.babraham.ac.uk/). Reads were trimmed for quality score and adaptor sequences were also removed. The high-quality reads (between 30 and 45 million) that passed quality filters were aligned to the human reference genome (GRCh38) using STAR aligner, allowing up to two mismatches in conjunction with the gene model from Ensemble 92. Reads were quantitated by counting the number of reads across exons. If no read count was present in 80% of samples that gene was excluded from the analysis. Differentially expressed genes were identified by combining two different approaches using two different algorithms. In order to call a gene "differentially expressed", a gene has to pass the false discovery rate (FDR)-adjusted $P < 0.05$ in DEseq[62], as well as in null model of hypothesis[63]. DesEQ2 performs a likelihood ratio test[64] that compares how well a gene count data fits a full model (with independent variable time) compared with a reduced model without those variables. The null model of hypothesis takes the average expression of groups into consideration. The gene list was further ranked using fold change criteria. To specifically study the effect of TFEB S142A and S142E on gene expression in two-group comparison, differential expression associated with each mutant was calculated in comparison with TFEB and vector control. Unless specified,

hierarchical clustering, principal component analysis, and statistical analysis were performed in R (http://www.r-project.org). The autophagy–lysosomal gene set was built based on various autophagy database such as Autophagy, human autophagy database (http://www.tanpaku.org/autophagy/), as well as published literature.

**GSEA**. GSEA analysis was used to study the enrichment of genes in different pathways[65]. Non-parametric GSEA was performed using GSEA 3.0 (Broad Institute, Cambridge, MA)[65]. This method ranks genes according to their relative difference in expression (Student's *t*-test) between two cell phenotypes. GSEA compares this ranked list of genes to a large collection of pathway data gene sets and assigns an enrichment score. If the gene is present in the dataset its score is increased, and if it is absent, the score is decreased. The enrichment statistics is the maximum derivation of running enrichment score from zero. The gene sets that significantly perform the random-class permutations are considered significant as detailed previously[66].

Public datasets (GSE20051[16], GSE50535[17], and GSE77940[18]) used for the GSEA meta-analysis in Supplementary Fig. 1d were downloaded from the Gene Expression Omnibus (GEO; http://www.ncbi.nlm.nih.gov/gds/). The expression of differentially induced/suppressed genes (FDR < 0.05) was validated by protein immunoblotting and/or qRT-PCR.

**Statistical analysis**. To ensure adequate power and decreased estimation error, we used large sample sizes and multiple independent repeats by independent investigators. In addition, multiple lines of experiments including different quantification methods were provided for consistent and mutually supportive results. The sample size was chosen according to well-established rules in the literature, as well as our ample previous research experience. Data are presented as mean ± SD. Statistical significance was calculated using the Student's *t*-test or one-way analysis of variance (ANOVA) test using GraphPad Prism 7.0 (GraphPad Software, Inc.), unless otherwise stated. A *P*-value of ≤0.05 was considered statistically significant.

**Reporting summary**. Further information on experimental design is available in the Nature Research Reporting Summary linked to this article.

## Data availability

Generated plasmids and cell lines are available from the corresponding author upon request. The RNA-seq data are deposited into Gene Expression Omnibus (GEO) with the accession code GSE122614. The source data underlying Figs. 1a, 1f, 2c–f, 2h, 3a, 3e, 3h, 3i, 4c, 4f, 5f, 5i, 6c, and 7e and Supplementary Figs. 2e, 2h, 4d, 5a, 5e, 6h, 7a, 7e, 7g, 7h–j, 9a, 10a, and 10d are provided as a Supplementary Fig. 13. All other data that support the findings of this study are available from the corresponding author on reasonable request.

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

## Acknowledgements

The authors wish to acknowledge Dr. A. Rodriguez for providing technical assistance for the experiments with electron microscopy. We thank all the members of the Liang laboratory for helpful discussion. We acknowledge the financial support from China Scholarship Council (CSC) of Shun Li for his PhD study at University of Southern California (USC). The authors wish to thank Dr. Martine Torres for her critical reading of the manuscript and editorial assistance. This work was supported by Melanoma Research Alliance (509218 to C.L.) and National Institutes of Health grants (R01 CA140964 and R01 ES029092 to C.L.).

## Author contributions

S.L. performed most experiments of this study and analyzed the data. Y.S., C.Q., H.G., G.-B.J., H.M., S.Z. and N.S. helped with animal experiments. V.P. conducted bioinformatics analysis. A.H. provided assistance in pathological analysis. Q.L., G.K.I., D.P., W.Y., K.M., M.,Y., O.A., Y.Y. and L.T. provided critical reagents and constructive discussion. C.L. designed the project, analyzed the data, and supervised all research.

## Additional information

**Competing interests:** The authors declare no competing interests.

