## [Peer Review File · Nature Communications]

Reviewers' comments:

Reviewer #1, Expertise: melanoma, BRAF (Remarks to the Author):

Overall the work elegantly dissects how the autophagy-lysosomal pathway is regulated in BRAFV600E melanoma cells and how current BRAFi therapy would impact on the metabolic signalling by reducing the efficacy of autophagy. This crucial observation, if confirmed, will drastically influence a potential approach of modulating the autophagy in combination with MAPK inhibitors for melanoma treatment. The authors also report that a reduced/impaired autophagy signalling (upon BRAFi treatment), eventually triggers another important signalling pathway - TGF β - which would make the melanoma even more aggressive and invasive.

This is a massive body of work with plenty of interesting and novel observations clarifying and, at the same time adding layer of complexity to the general knowledge about autophagy regulation in BRAFV600E melanoma. The data supports most of the claims discussed in the manuscript although a number of issues have been identified (in particular the autophagy data, with specific point on the LC3I isoform & the BRAFi resistance part, which is not well developed) which should be addressed, to further consolidate the strength of this report.

Minor points

- 1) At page 5 (line 120): ...Blockade of BRAFi-induced transcriptional output of autophagy-lysosomal activation....perhaps the authors meant "blockade of BRAF-induced transcriptional"
- 2) At page 29 (line 762): in legend of figure 1A-B should be mentioned for how long cells (A375 and MeWo) were exposed with PLX4720.
- 3) The authors suggest at page 7 (lane169) that faster-migrating TFEB isoform displayed in top panel of figure 2C could be the result of reduced phosphorylation. This reviewer disagree as the molecular size differences of the 2 isoforms is quite significant, at least in this figure and as well in figure 2E. Following results later in the manuscript supports this statement, although no evidence is provided to confirm that this molecular weight changes in TFEB are specifically due to a de-phosphorylation (or activation of phosphatases) . The same TFEB isoform in a much less amount is also expressed into the cytoplasmic fraction (top panel of figure 2C).
- 4) It is remarkable how in most of IF images of A375 cells TFEB (with DMSO) is homogeneously distributed into the cytoplasm, whereas in fig 3B its expression (with DMSO) mainly localize at the lysosomes!!
- 5) At page 13 (lane 306) Suppl. fig. 5f (instead of 6f).

Major points

6) Interpretation of Autophagy efficacy, LC3 isoforms. In most of the Western blots data (fig.1f; 3i; S4d; S5a; S7a; S7e; S9d; S10e-f) of LC3 to confirm the autophagy activity rather than showing an increasing amount of LC3 II (lipidated isoform) is often reported a decreasing expression of LC3I (untransformed LC3 isoform). The ratio between the 2 isoforms overall is increasing, yet this common feature (changes of the expression of LC3I isoform) might unveil a novel mechanistic insight specifically induced by BRAF inhibition. The most emblematic evidence of this trend is shown in suppl. figure 5a, where upon expression of non-phosphorylatable TFEB S142A LC3 ratio increased (p62 decreased) in xenografted NOD/SCID mice with A375 tumours when treated with PLX4720. Exactly the opposite is described with expression of phospho-mimetic TFEB S142E, which the authors claimed overall decreased the autophagy signalling (decreased LC3 ratio, increase p62 expression). By carefully examining the western blot, LC3II expression is pretty stable among all

the samples, the only differences are in the changes of the untransformed version of LC3 (LC3I). Could the authors provide an explanation about this relevant issue?

7) The interpretation of apoptosis from immunohistochemistry in figure 5d is disputable. I do not observe a significant increase in Caspase 3 upon TFEBS142E expression nor an increase in PARP cleaved expression in Western Blot in suppl. fig. 5A. Moreover, a reduced number of mitochondria and increased ROS activity might reflect more likely a mitophagy scenario and an overall genomic instability situation rather than an apoptotic one.

8) At page 16 (lane 369) the authors make this statement: “..TFEB might regulate TGF- β protein turnover through the autophagy-lysosomal degradation pathway”. Although they have provided a lot of evidence, unfortunately they have also clearly shown that TGF- β feed forward loop does not control transcriptionally the autophagy-lysosomal degradation pathway. The protein turnover is not a demonstrated mechanism in the manuscript it is just a speculative suggestion of the authors. Could the author provide and demonstrate an alternative mechanism (microRNA?).

9) LC3 I and II expression has not been included in both figures 5i and Suppl. S6f. This is a crucial control for demonstrating the efficacy of Bafilomycin A1 treatment on the cellular system. Moreover, by adding this important control the authors could clearly provide info about the ability to specifically stabilized LC3 II isoform in their cellular system. Finally, another crucial control such as Beclin1, should have been included in this figure to confirm the efficacy of BECN1 SHRNA silencing.

10) In figure 6a the authors claim that “..tumor metastasis correlated with suppressed autophagy-lysosomal function” (by only showing p62 immunostaining): why the expression of vector alone would increase tumor formation (tumor size and number of metastasis) and p62 staining? More importantly, what about LC3 staining in those tumors, why the authors have decided not to show it?

11) The BRAFi resistance. In general, this is probably the weakest part of the story. According to the data provided once melanoma cells acquire BRAFi resistance TFEB remains constitutively phosphorylated, preventing its relocation into the nucleus upon further treatment with PLX4720, and the autophagy signalling no longer respond to further BRAFi activation. In this scenario few things are not completely clear.

A. From the very beginning of the story, it has not been shown (a part from its intracellular localization) how TFEB total expression is affected by BRAFi treatment and how this could further change when melanoma cells become PLX-resistant.

B. In a very elegant report on this journal has been well characterized how TFEB intracellular location is also dependent on CRM1-mediated nuclear export [Nature Communications (2018) 9:2685]. During acquisition of BRAFi resistance could this mechanism have been impaired, preventing TFEB nuclear re-localization, on top of his acquired insensitivity of being de-phosphorylated?

C. In this BRAFi resistance what is also happening to the other arm of the transcriptional regulator of the autophagy-lysosomal machinery, ZKSCAN3? Does its expression/ location changes upon BRAFi resistance?

Reviewer #3, Expertise: autophagy and cancer
(Remarks to the Author):

The manuscript by Li et al describes a role for mutant Braf V600E and ERK dependent phosphorylation of TFEB in regulation of autophagy and lysosome biogenesis in the context of

melanoma growth and metastasis. The authors show that treatment of melanoma cell lines harboring V600E mutations with a BRAF inhibitor leads to nuclear translocation and activation of TFEB in addition to simultaneous nuclear exit of ZKSCAN3 (a negative transcriptional regulator of autophagy). The shuttling of TFEB and ZKSCAN3 in and out of the nucleus respectively is mediated by changes in their phosphorylation status. The authors provide compelling evidence that BRAFi treatment suppressed ERK dependent phosphorylation of TFEB at S142 (sending it into the nucleus) and stimulated p38/JNK dependent phosphorylation of ZKSCAN3 (sending it out of the nucleus). The authors also provide some evidence that TFEB localization is independent of mTOR activation status – however this data is less convincing (see comments below). The consequences of this regulation is presented in the second half of the manuscript where the authors provide evidence supporting a tumor promoting role for inactive TFEB (S142E) and a tumor suppressive role for activated TFEB (S142A) in melanoma tumorigenesis. Mechanistically the authors propose that TFEB suppression and the corresponding decrease in autophagy/lysosome function blocks autophagy mediated degradation of TGFbeta and promotes EMT and increased primary and metastatic colonization and growth.

A substantial amount of solid data is presented in support of the authors claims that mutant BRAF inhibits TFEB to promote tumorigenesis. However some caveats with the chosen systems used in study and additional concerns regarding some of the authors claims are detailed below.

1. The authors use 2 BRAF mutant cell lines in their study – A375 (homozygous V600E Braf mutant), and G361 (heterozygous V600E Braf mutant). Are the effects of mut BRAF and BRAFi on TFEB localization broadly applicable in a wider set of Braf mutant melanoma vs WT or NRAS mutant melanoma cell lines?

2. the authors should use antibodies against p-S142 to definitely show regulation by ERK

3. S142 is also phosphorylated by mTORC1 in a Rag GTPase dependent manner. The authors show that a constitutively active mTOR (E2419K) does not prevent nuclear shuttling of TFEB in response to BRAFi treatment. Are the mTOR and ERK CA mutants expressed at comparable levels? The authors should use mTOR related perturbations that address the importance of Rag mediated activation in regulation of TFEB phosphorylation and localization. For instance - use a constitutively active RagGTPase (Q99), knockdown of GATOR1 components (renders RagA constitutively active). Similarly Raptor knockdown does not appear to override constitutively active ERK in maintaining TFEB-GFP in the cytoplasm. In this experiment a western blot showing the efficiency of Raptor KD should be shown and quantification of nuclear vs cytoplasmic localization should be presented.

3. The in vitro and in vivo tumorigenicity assays are all conducted using A375 cells overexpressing TFEB mutants, which is a weakness. In some instances the levels of overexpression are not equal for the individual mutants (see figure 7e, supplemental figure 5a). What would be the effect of overexpressing these same constructs in a control cells line (ei - MeWo -Braf WT cells) on tumor growth? Can the authors use CRISPR editing of the TFEB locus to introduce the point mutants and confirm a subset of their findings?

4. The authors should more rigorously establish whether the effects of TFEB suppression on EMT and tumor growth are indeed autophagy/lysosome dependent or independent. The authors suggest that EMT induction and aggressive tumor growth following expression of inactive TFEB is due to a block in autophagy mediated degradation of TGFbeta. How is specificity for this cargo established? How relevant is this cargo relative to other proteins that are stabilized following TFEB inactivation? Wouldn't autophagy blockage via ATG knockdown, chloroquine treatment (independent of TFEB) have the same effect? Have the authors tested this? Do Braf mutant melanoma cells have overall higher TGFbeta

5. what are the baseline levels of autophagy, TFEB nuclear status, EMT in A375 versus A375R cells ? Are A375R cells more or less aggressive than the parental cell line?

Finally, the authors discuss that the role of autophagy in tumor growth is context dependent and potentially tissue specific – they support this idea with several references. However, their findings in melanoma contrast several reports in the same tissue type showing that autophagy activation is required for melanoma growth – eg. ATG5/7 knockout in Braf mutant melanoma mouse model suppresses tumor growth: Xie X et al Cancer Discov 2015; hydroxychloroquine is associated with tumor inhibition in melanoma patient clinical trials: Rangwala R et al Autophagy 2014; lysosomal inhibition inhibits melanoma growth: Rebecca VW et al Cancer Discov 2017; patients with autophagy induction following Braf inhibitor treatment showed reduced response to therapy and poor prognosis: Ma et al J Clin Invest. These studies all suggest that autophagy activation is required for tumor growth and portend worse overall outcomes. How do the authors reconcile their data with these previously published findings in the same cancer type?

Minor points:

1. The autoradiograph in figure 3d related to the kinase assay is not convincing and should be replaced.

2. p-ERK levels should be shown in fig S9d

3. the concentration of PLX4720 used and duration of treatment should be indicated in the figure legend for each experiment (eg. missing in fig S9c)

RESPONSE TO REVIEWERS

We are truly appreciative of the reviewer's constructive and insightful comments, according to which the manuscript has been carefully and rigorously revised. We hope the new version of our manuscript is now appropriately suited for publication in *Nature Communications*. A detailed response to the Reviewer's critiques and a description of the new experiments (in *italic*) follow:

Point-by-point Response to Reviewer #1

Reviewer 1 stated that “the work elegantly dissects how the autophagy-lysosomal pathway is regulated in BRAFV600E melanoma cells and how current BRAFi therapy would impact on the metabolic signaling by reducing the efficacy of autophagy. This crucial observation, if confirmed, will drastically influence a potential approach of modulating the autophagy in combination with MAPK inhibitors for melanoma treatment. The authors also report that a reduced/impaired autophagy signaling (upon BRAFi treatment), eventually triggers another important signaling pathway -TGFβ- which would make the melanoma even more aggressive and invasive. This is a massive body of work with plenty of interesting and novel observations clarifying and, at the same time adding layer of complexity to the general knowledge about autophagy regulation in BRAFV600E melanoma.

Response: *We truly appreciate the reviewer's enthusiasm and great comments. Rigorous efforts have been taken to address the deficits noted by the Reviewer, as detailed below.*

Minor points:

1) At page 5 (line 120): ...Blockade of BRAFi-induced transcriptional output of autophagy-lysosomal activation. ...perhaps the authors meant “blockade of BRAF-induced transcriptional”

Response: *Thank the reviewer for this comment. Please kindly note that TFEB-mediated transcriptional activation of autophagy-lysosomal function is suppressed by oncogenic BRAF^{V600E} signaling through ERK-mediated TFEB phosphorylation in BRAF^{V600E} melanoma. BRAF inhibitor (BRAFi) inactivates ERK, thus activating TFEB function as a transcriptional factor in the autophagy-lysosome pathway. For clarity, the original sentence has been re-phrased to “Blockade of BRAFi-induced transcriptional activation of autophagy-lysosomal function in BRAF-mutant melanoma causes increased tumor progression, EMT-associated transdifferentiation, and partial resistance to BRAFi-therapy”.*

2) At page 29 (line 762): in legend of figure 1A-B should be mentioned for how long cells (A375 and MeWo) were exposed with PLX4720.

Response: *Thank the reviewer for this comment. The legend has been updated with the exposure time of PLX4720.*

3) The authors suggest at page 7 (line 169) that faster-migrating TFEB isoform displayed in top panel of figure 2C could be the result of reduced phosphorylation. This reviewer disagree as the molecular size differences of the 2 isoforms is quite significant, at least in this figure and as well in figure 2E. Following results later in the manuscript supports this statement, although no evidence is provided to confirm that this molecular weight changes in TFEB are specifically due to a de-phosphorylation (or activation of phosphatases) . The same TFEB isoform in a much less amount is also expressed into the cytoplasmic fraction (top panel of figure 2C).

Response: We have carefully repeated the TFEB western blot (WB) experiments in Figure 2c and 2e and provided new representative data in the revised manuscript, which is now consistent with other TFEB WB results throughout the manuscript and the literature as well. We have also used p-S142-specific TFEB antibody and confirmed that the faster-migrating TFEB form lost S142 phosphorylation (attached here for your reference; also refer to Fig. 3a). The original TFEB WB results in Figure 2c and 2e, which showed faster mobility of de-phosphorylated TFEB than expected, is most likely due to an altered PAGE gel condition and the buffer system used. Specifically, a 4-20% gradient PAGE gel (GenScript M42012) with Tris-MOPS running buffer was used in original Figures 2c and 2e. We have repeated the experiments using the consistent setting (i.e., non-gradient SDS-PAGE with a Tris-Glycine buffer). Furthermore, the trivial amounts of the faster-migrating TFEB form in the cytoplasmic fraction in the original top panel of Figure 2c was due to contamination during fractionation of the cytoplasmic and nuclear fractions. This experiment has been carefully repeated with more gentle treatment of cells with fractionation buffer. The new data is provided in the revised manuscript (Refer to Fig. 2c) and also attached here for your reference.

Fig. Immunoblots for TFEB and S142-phosphorylated TFEB in the cytoplasmic and nuclear fractions of A375 cells treated with DMSO or PLX4720 (1 μM, 12 hr). Lamin B1 serves as the control for the nuclear fractions, while LAMP1 and tubulin are the controls for the cytoplasmic fractions. Also refer to Fig. 2c.

4) It is remarkable how in most of IF images of A375 cells TFEB (with DMSO) is homogeneously distributed into the cytoplasm, whereas in fig 3B its expression (with DMSO) mainly localize at the lysosomes!!

Response: Thank the reviewer for this comment. Endogenous TFEB is mainly localized to the cytoplasm, but also with focal concentrations at the juxtannuclear region associated with lysosomes under basal conditions (DMSO), as reflected in most of our confocal images of TFEB in our work and in the literature^{1,2}. Unfortunately, while scaling down multi-cell images to fit into the restricted figure frame, this fine pattern might be blurred a bit. We therefore attached an enlarged representative image of TFEB for your reference. We apologize for the less representative image in Fig. 3b of the original manuscript. We have carefully repeated this experiment and provided more representative data in the same figure.

Fig. Confocal microscopy analysis of endogenous TFEB distribution relative to the lysosome compartment (LAMP1-labelled). Please also refer to **Fig. 3b** in the revised manuscript. Scale bars. 10 μm.

5) At page 13 (lane 306) Suppl. fig. 5f (instead of 6f).

Response: We thank the reviewer for pointing out this error. It has been fixed in the revised manuscript.

Major points:

6) Interpretation of Autophagy efficacy, LC3 isoforms. In most of the Western blots data (fig.1f;3i; S4d; S5a; S7a; S7e; S9d; S10e-f) of LC3 to confirm the autophagy activity rather than showing an increasing amount of LC3 II (lipidated isoform) is often reported a decreasing expression of LC3I

(untransformed LC3 isoform). The ratio between the 2 isoforms overall is increasing, yet this common feature (changes of the expression of LC3I isoform) might unveil a novel mechanistic insight specifically induced by BRAF inhibition. The most emblematic evidence of this trend is shown in suppl. figure 5a, where upon expression of non-phosphorylatable TFEB^{S142A} LC3 ratio increased (p62 decreased) in xenografted NOD/SCID mice with A375 tumours when treated with PLX4720. Exactly the opposite is described with expression of phospho-mimetic TFEB^{S142E}, which the authors claimed overall decreased the autophagy signalling (decreased LC3 ratio, increase p62 expression). By carefully examining the western blot, LC3II expression is pretty stable among all the samples, the only differences are in the changes of the untransformed version of LC3 (LC3I). Could the authors provide an explanation about this relevant issue?

Response: Thank the Reviewer for this critical comment. To further justify this concern per Reviewer's suggestion, we rigorously re-conducted all of the LC3 Western blot experiments in Fig. 1f, Fig. 3i, Fig. 6c, Fig.7e, and Figures S4d, S5a, S8a (original S7a), S9a, S10d (original S9d), and 11e-f (original 10e-f), using a different LC3 antibody (Cell signaling, #2775S) generated against the N-terminal peptide of LC3. All results showed a consistent and reverse correlation of increased amount of LC3-II with decreased amount of LC3-I as well as with decreased levels of p62, and vice versa. Although a similar pattern was also observed in the original manuscript by using the LC3 antibody generated against human recombinant LC3 (CAC-CTB-LC3-2-IC, Cosmo Bio), this antibody tends to show less affinity to LC3-II than LC3-I, as noted by the reviewer. Similar phenomenon has also been reported in the literature such that antibody derived from N-terminal peptide of LC3 showed higher sensitivity for detection of LC3-II than of LC3-I (Mizushima and Yoshimori, *Autophagy* 3:6, 542-545. 2007)³. It is therefore speculated that some conformational change might occur at the N-terminus of LC3 after PE-conjugation, resulting in exposure of an antibody-reactive epitope and better detection (Mizushima and Yoshimori, *Autophagy* 2007)³. We have supplemented all of the LC3 Western blot data with corresponding re-quantification in the revised manuscript, and also attached one example of Figure S5a for the reviewer's reference.

7) The interpretation of apoptosis from immunohistochemistry in figure 5d is disputable. I do not observe a significant increase in Caspase 3 upon TFEB^{S142E} expression nor an increase in PARP cleaved expression in Western Blot in suppl. fig. 5A. Moreover, a reduced number of mitochondria and increased ROS activity might reflect more likely a mitophagy scenario and an overall genomic instability situation rather than an apoptotic one.

Response: We highly agree with the Reviewer that tumors expressing TFEB^{S142E}, which showed defective autophagy, particularly mitophagy, exhibited increased oxidative damage and overall genomic instability, as exemplified by high levels of γ -H2AX (a DNA double-strand break marker⁴) in Fig. S5a of the original manuscript. Although the relationship between autophagy, apoptosis, and

Fig. WB analysis of LC3 in the indicated xenograft tumor genotypes (three randomly chosen samples per group). Also refer to **Fig. S5a**.

Fig. WB analysis of PARP and cleaved caspase 3 (top panels) and IHC analysis of cleaved caspase 3 (bottom panels) in the indicated xenograft tumor genotypes. Refer to **Fig. 5d** and **S5a**.

genomic instability is complex and context-dependent, autophagy deficit and/or genomic instability can activate the apoptotic program^{5,6}. To clarify this issue, we carefully re-examined the immunohistochemical (IHC) staining of active caspase-3 in Fig. 5d and repeated the immunoblotting experiments for both cleaved caspase-3 and PARP in Fig. S5a. As also shown in the attached figures, expression of TFEB^{S142E} is associated with increased IHC staining of active caspase-3 and increased levels of active caspase-3 and PARP cleavage in xenograft tumors, while opposite manner was detected upon TFEB^{S142A} expression. More representative data has been provided in Fig. 5d and Fig. S5a of the revised manuscript after rigorous repeat and experimental optimization.

8) At page 16 (lane 369) the authors make this statement: “..TFEB might regulate TGF- β protein turnover through the autophagy-lysosomal degradation pathway”. Although they have provided a lot of evidence, unfortunately they have also clearly shown that TGF- β feed forward loop does not control transcriptionally the autophagy-lysosomal degradation pathway. The protein turnover is not a demonstrated mechanism in the manuscript it is just a speculative suggestion of the authors. Could the author provide and demonstrate an alternative mechanism (microRNA?).

Response: We thank the reviewer for this critical comment. We have now included new datasets in Fig. S7a to S7h of the revised manuscript to demonstrate that TFEB-mediated activation of autophagy-lysosome function regulates TGF- β protein turnover (also attached here for your reference). Briefly, the increase in TGF- β protein without a concomitant increase in mRNA level upon TFEB^{S142E} expression suggested that TGF- β protein is subjected to post-translational regulation. Indeed, we found that inhibition of autophagic flux by the lysosomotropic chloroquine (CQ), or silencing autophagy essential genes Beclin1 and Atg5, markedly increased TGF- β proteins in A375 cells without affecting the mRNA levels (Fig. S6a-c), indicating a steady-state turnover of TGF- β through the autophagy-lysosomal pathway. Notably, TGF- β protein contains potential LC3-interacting region (LIR) motifs (Fig. S7d), a core consensus sequence of (WFY)XX(ILV)⁷⁻⁹. We therefore tested whether

Fig. S7 (a,b) Western blot analysis of TGF- β and autophagy markers in A375 cells treated with chloroquine (CQ; 50 μ M) for indicated time (a) or in A375 cells with shRNA-mediated depletion of Atg5 and Beclin1 (b). Actin serves as a loading control. (c) Quantitative RT-PCR analysis of TGF- β mRNA expression in cells in (a) and (b). (e) Expression of TFEB^{S142A} promotes interaction between endogenous pro-TGF- β 1 and LC3 in A375 cells, which is inhibited by 3-MA (100 nM, 24 hr). Also refer to **Fig. S7a-S7h**.

LC3 binds to TGF- β targeted for selective turnover in the lysosomes. As expected, endogenous interaction between pro-TGF- β and LC3 was readily detected by co-immunoprecipitation in A375 cells even under basal condition (Fig. S7e). Expression of TFEB^{S142A} induced autophagy activation and concomitantly promoted pro-TGF- β association with LC3. Treatment of cells with 3-methyladenine (3-MA), a class III PI3K inhibitor that blocks autophagosome biogenesis, ablated the effect of TFEB^{S142A}, resulting in TGF- β accumulation (Fig. S7e). In accord, a significant quantity of

pro-TGF- β was present in LC3-labeled autophagosomes and LAMP1-labeled lysosomes upon TFEB^{S142A} expression, whereas much less was found in TFEB^{S142E}-expressing cells (Fig. S7f). We also found that disruption of cytoplasmic autophagy by depletion of Beclin1¹⁰ or by Bafilomycin A1 (BafA1)¹¹ restored TGF- β protein levels that were suppressed by TFEB^{S142A} in A375 cells (Fig. S7h). Together, these results indicate that TGF- β proteins are selective cargo of autophagy and that TFEB acts on TGF- β protein turnover through the autophagy-lysosomal pathway.

Although we did not observe clear transcriptional control of autophagy-lysosome function by TGF- β in our experimental settings, we did show that TGF- β signaling contributes to the tumor de-differentiation and drug resistance of BRAF^{V600E} melanoma. This was observed when treating cells with TGF- β inhibitor abolished the effect of TFEB^{S142E} in promoting EMT and drug resistance, while supplementing more TGF- β rendered TFEB^{S142A} incapable of suppressing EMT and drug resistance. To avoid confusion, we have deleted “feed forward loop” from the main text and rephrased it as “BRAF^{V600E}-dependent TFEB phosphorylation and consequent TGF- β activation enhances BRAF^{V600E}-driven tumor aggressiveness”. Please also refer to new Fig. S7 and corresponding text for details.

9) LC3 I and II expression has not been included in both figures 5i and Suppl. S6f. This is a crucial control for demonstrating the efficacy of Bafilomycin A1 treatment on the cellular system. Moreover, by adding this important control the authors could clearly provide info about the ability to specifically stabilized LC3 II isoform in their cellular system. Finally, another crucial control such as Beclin1, should have been included in this figure to confirm the efficacy of BECN1 SHRNA silencing.

Response: *The immunoblotting of LC3 and Beclin1 have been included in Fig. 5i and Fig. S7h (original Fig. S6f) of the revised manuscript. As expected, efficient Beclin1 depletion was achieved, which inhibited autophagy even in the presence of TFEB^{S142A} expression. BafA1 treatment blocked autophagic flux, leading to accumulation of LC3-II isoform.*

10) In figure 6a the authors claim that “..tumor metastasis correlated with suppressed autophagy-lysosomal function” (by only showing p62 immunostaining): why the expression of vector alone would increase tumor formation (tumor size and number of metastasis) and p62 staining? More importantly, what about LC3 staining in those tumors, why the authors have decided not to show it?

Response: *In Figure 6a, to evaluate the impact of TFEB (activation vs. inactivation) on BRAF^{V600E}-associated tumor metastatic potential in an immune-competent background, a mouse melanoma B16-F10 syngeneic model¹² was used. The B16-F10 cells were engineered to stably express oncogenic BRAF^{V600E} along with vector control or TFEB (WT or mutant). This B16-F10 cell line was originally derived from mouse melanoma¹³, therefore it has tumor-initiating ability and can give rise to tumor formation even at the background level (vector alone). We have now provided the immunoblotting results of LC3 and p62 in tumor metastases in Fig. 6c (also attached here for the Reviewer’s reference), which is consistent with IHC p62 staining in mouse tissues, and with immunoblotting data of corresponding B16-F10 stable cell lines (original Fig. S7a and now Fig. S8a), from which the tumors were reconstituted. Unlike LC3 immunoblotting, which can differentiate the LC3-II isoform from the LC3-I isoform by mobility shift on SDS-PAGE gel, IHC staining of LC3 virtually labels all LC3 proteins regardless of their isoforms, thus we choose to use WB of LC3 and p62 to further justify the*

Fig. 6c

Fig. 6c. Immunoblot analyses of EMT, TGF- β , LC3 conversion, and p62 in indicated lung metastases (two randomly chosen samples per group). Actin served as a loading control.

correlation between tumor metastasis and autophagy status in mice in vivo. The new data has been provided in Fig. 6 and S8 with the corresponding text revised.

11) The BRAFi resistance. In general, this is probably the weakest part of the story. According to the data provided once melanoma cells acquire BRAFi resistance TFEB remains constitutively phosphorylated, preventing its relocation into the nucleus upon further treatment with PLX4720, and the autophagy signalling no longer respond to further BRAFi activation. In this scenario few things are not completely clear.

Response: Thank the Reviewer for this comment. Previous studies have claimed that inhibition of BRAFi-induced autophagic response can enhance the response of melanoma cells to BRAFi, but they suffer from technical limitations such as the use of chloroquine (CQ) as an autophagy-selective inhibitor while ignoring its anti-tumor effects through autophagy-independent mechanisms. Moreover, cancer trials that combine autophagy inhibitors with BRAF inhibitors have so far shown incomplete and short-term responses in melanoma patients^{14,15}, reflecting a serious gap in our understanding of the relationship between oncogenic signaling and autophagy in melanoma. In this work, we used the mechanism-directed approach to ablate BRAFi-induced autophagy (note: BRAFi induces autophagy by TFEB activation) by expressing a TFEB phosphomimetic mutant (TFEB^{S142E}). We found that blocking BRAFi-induced TFEB activation and autophagy-lysosome function adversely (rather than favorably) affects tumor response to PLX4720 treatment as shown in Fig. 8. We have provided additional data to further justify that PLX4720 activate TFEB but did not affect TFEB expression level (detailed below in A), and that this regulation of TFEB is independent of CRM1 (detailed below in B), and that in BRAFi-resistant A375^R cells, ZKSCAN3 failed to translocate to the cytoplasm upon PLX4720 treatment, although its overall expression remained unchanged (detailed below in C).

A. From the very beginning of the story, it has not been shown (a part from its intracellular localization) how TFEB total expression is affected by BRAFi treatment and how this could further change when melanoma cells become PLX-resistant.

Response: We would like to kindly note that PLX4720 treatment, which inhibited ERK-mediated TFEB phosphorylation, promoted TFEB nuclear translocation and activation, had minimum effect on the overall levels of TFEB, as indicated in the Western Blots of TFEB in Fig. 2c, 2d, 2e and in the qRT-PCR analysis of TFEB in Fig. 1e and Fig.S1c. Moreover, TFEB expression remained unchanged even when melanoma cells become PLX4720-resistant, as shown in the Western Blots of TFEB of Fig. S9a (original Fig. S8a). Notably, in PLX4720-resistant A375^R melanoma cells, TFEB failed to translocate to the nucleus and constitutively interacted with 14-3-3 in the cytoplasm regardless of PLX4720 treatment (now Fig. S9a). This is largely due to the constitutive ERK activation and thereof TFEB inactivation in drug-resistant melanoma cells, regardless of BRAF inhibition. Similarly, qRT-PCR analyses of TFEB in both drug-sensitive and -resistant A375 cells revealed no difference before or after PLX4720 treatment (see the attached figure). We have also revised the manuscript to further emphasize this point.

Fig. Real-time qRT-PCR of TFEB expression in PLX4720-sensitive A375 and PLX4720-resistant A375^R cells in the presence/absence of PLX4720 treatment.

B. In a very elegant report on this journal has been well characterized how TFEB intracellular location is also dependent on CRM1-mediated nuclear export [Nature Communications (2018) 9:2685]. During acquisition of BRAFi resistance could this mechanism have been impaired, preventing TFEB nuclear re-localization, on top of his acquired insensitivity of being de-phosphorylated?

Response: We thank the Reviewer for this valuable comment. Napolitano et al., (Nat. Commun. 2018) reported that nutrients and mTORC1 signaling regulate TFEB nuclear export in a CRM1-dependent

manner, as *CRM1* silencing or *leptomycin B* treatment prevents cytosolic re-localization of nuclear TFEB, leading to nuclear accumulation of TFEB even in a phosphorylated state and in nutrient-rich condition. To further investigate whether PLX4720-induced TFEB de-phosphorylation/activation (vs. ERK(CA)-mediated TFEB S142 phosphorylation/inactivation) also involves this CRM1 mechanism, we depleted CRM1 from ERK(CA) cells, which are insensitive to PLX4720 treatment. As shown in Fig. 3e

(also refer to the attached figure), no nuclear retention of TFEB(WT) was detected in ERK(CA)-expressing cells upon CRM1 knockdown. Furthermore, upon ERK activation, S142A TFEB mutant constitutively translocated to the nucleus, while S211A TFEB mutant constitutively resided in the cytoplasm, regardless of CRM1 expression. Analogous results were also obtained when cells were treated with leptomycin B, a known CRM1 inhibitor (Fig. 3e; refer to the attached figure). These results indicate that ERK-mediated S142 phosphorylation is a predominant mechanism to sequester TFEB in the cytoplasm, overriding the effect of CRM1, in BRAF^{V600E} melanoma cells. In fact, we have shown in Fig. 3f and 3h that TFEB^{S142E} binds to 14-3-3 and localizes to the lysosome regardless of PLX4720 treatment. Additionally, as noted by Napolitano et al., (Nat. Commun. 2018), CRM1-mediated TFEB nuclear export is regulated by mTOR-dependent phosphorylation. However, constitutive activation of mTORC1 by either overexpression of mTORC1 (CA), its upstream activator RagB GTPase (Q99L), or by silencing the mTORC1 inhibitor DEPDC5, cannot ablate PLX4720-induced TFEB nuclear translocation (Fig. 2f; refer to the attached figure). Conversely, inactivation of mTORC1 by depletion of Raptor cannot prevent ERK(CA)-mediated cytoplasmic retention of TFEB (Fig. 3e; refer to the attached figure). These results, together with the data of TFEB point mutants (S142A/E vs. S211A/E), demonstrated a mTOR-independent regulation of TFEB by ERK in BRAF^{V600E} melanoma cells. We have supplemented new data in Fig. 2f, Fig. 3e, and Fig.S2i and revised the manuscript accordingly.

Fig. 3e

Fig. 2f

Fig. 3e Representative confocal images of subcellular distribution of WT and S142A and S211A TFEB mutants in A375 cells stably expressing ERK(CA) in the presence or absence of PLX4720 (1 μM, 12 hr), Leptomycin B (20 nM, 2 hr) or treated with Raptor-specific or CRM1-specific shRNA.

Fig. 2f Representative images of nuclear localization of endogenous TFEB (green) in A375 cells stably expressing the constitutive active (CA) form of mTORC1 (E2419K), RagB GTPase (Q99L), or ERK (R67S/D321N), or expressing DEPDC5-specific shRNA, with or without the treatment of PLX4720 (1 μM, 12 hr; 2nd row), or ERK inhibitor FR180204 (ERKi, 10 μM, 24 hr; 3rd row).

C. In this BRAFi resistance what is also happening to the other arm of the transcriptional regulator of the autophagy-lysosomal machinery, ZKSCAN3? Does its expression/ location changes upon BRAFi resistance?

Response: To address the reviewer's concern, we first examined ZKSCAN3 expression and location in PLX4720-resistant isogenic A375^R melanoma cells. As observed with TFEB, which remained S142 phosphorylated in the cytoplasm regardless of PLX4720 treatment, ZKSCAN3 was unresponsive to PLX4720-induced cytoplasmic translocation and its expression remained unchanged (Fig. S9a and S9b; also refer to the attached Fig. a). This is because in A375^R melanoma cells with a second NRAS(Q61K) mutation (upstream of BRAF), ERK remained active while JNK2 remained suppressed (refer to Fig. S9a WB data), leading to pertinent inactivation of TFEB transcriptional activator and activation of ZKSCAN3 transcriptional suppressor.

We also examined the status (expression/location) of ZKSCAN3 in A375 cells expressing TFEB

phosphomimetic ($TFEB^{S142E}$), which were resistant to PLX4720-induced TFEB activation. As expected, TFEB inactivation had no impact on the overall expression of ZKSCAN3 (now Fig. S10a), nor did it affect PLX4720-induced ZKSCAN3 cytoplasmic translocation/inactivation (refer to the attached figure b). Still, overall autophagy-lysosome function remained

suppressed due to the inactivation of the key transcriptional activator TFEB despite the cytoplasmic relocation of ZKSCAN3 by PLX4720 in $TFEB^{S142E}$ -complemented $TFEB^{KD}$ cells, (Fig. S10a). These data are in strong support of two distinct mechanisms of autophagy-lysosome activation by BRAFi, one through the ERK-TFEB axis and the other through the JNK2-ZKSCAN3 axis, which function synergistically to regulate the net production of autophagy-lysosome-relevant factors and tumor suppression, as exemplified in Fig. 4 and Fig. S5f-h. We have supplemented new data in Fig. S9a, S9b, and S10a and revised the manuscript accordingly.

Finally, we thank the Reviewer for his/her suggestions that have led to a much-improved manuscript.

Fig. (a) Confocal images showing subcellular distribution of ZKSCAN3 in A375 and A375^R cells in response to DMSO or PLX4720 treatment. The percentage of cells with cytoplasmic distribution of ZKSCAN3 is quantified (right panel). **(b)** Subcellular distribution of TFEB and ZKSCAN3 in TFEB knockdown (TFEB sh) A375 cells with re-expression of $TFEB^{S142E}$. Note TFEB-independent cytoplasmic translocation of ZKSCAN3 in response to PLX4720. Scale bars, 10 μ m. Also refer to **Figure S9a,b**.

Point-by-point Response to Reviewer #3

Reviewer 3 stated that “The manuscript by Li et al describes a role for mutant Braf V600E and ERK dependent phosphorylation of TFEB in regulation of autophagy and lysosome biogenesis in the context of melanoma growth and metastasis. A substantial amount of solid data is presented in support of the authors claims that mutant BRAF inhibits TFEB to promote tumorigenesis. However some caveats with the chosen systems used in study and additional concerns regarding some of the authors claims are detailed below.”

Response: We are truly appreciative of the reviewer’s enthusiasm and great suggestions, which were very helpful in revising the manuscript. A detailed response to the Reviewer’s critiques and a description of the new experiments follow:

Critique

1. The authors use 2 BRAF mutant cell lines in their study – A375 (homozygous V600E Braf mutant), and G361 (heterozygous V600E Braf mutant). Are the effects of mut BRAF and BRAFi on TFEB localization broadly applicable in a wider set of Braf mutant melanoma vs WT or NRAS mutant melanoma cell lines?

Response: We thank the Reviewer for this suggestion. We have examined TFEB localization in another BRAF^{V600E}-positive melanoma cell line SK-MEL-5, and found that PLX4720 induced nuclear translocation of TFEB similar to those observed in A375 and G361 cells (Fig. S2d; also refer to the attached figures). By contrast, no nuclear translocation was detected for TFEB in NRAS mutant (Q61R) SK-MEL-2 melanoma cells that contain WT BRAF (Fig. S2d; also refer to the attached figures). Notably, TFEB nuclear translocation was also observed in PLX4720-treated HT29 colon carcinoma cells bearing the BRAF^{V600E} mutation, suggesting this is likely a BRAF^{V600E}-specific event (Fig. S2d; refer to the attached figures). We have supplemented new data of Fig. S2d and revised the manuscript accordingly to include this critical point.

Fig. Confocal microscopy analyses of the subcellular distribution of endogenous TFEB in PLX4720 (1 μ M, 12 hr)-treated SK-MEL-5 (BRAF^{V600E}-positive) and SK-MEL-2 (NRAS^{Q61R}, BRAF^{WT}) melanoma cells, and HT29 (BRAF^{V600E}-positive) colon cancer cells.

2. the authors should use antibodies against p-S142 to definitely show regulation by ERK.

Response: We have now included the new data of TFEB phosphorylation using the S142 phospho-specific antibody in Fig. 3a (also attached here for reference). Using the S142 phospho-specific antibody, we found that TFEB S142 phosphorylation is highly increased in A375 cells compared with MeWo cells (Fig. 3a). Treatment of cells with PLX4720 completely abolished the phosphorylation of TFEB at Ser142 (Fig. 3a), in concordance with the result of in vitro kinase assay.

Fig. 3a. PLX4720 inhibits TFEB interaction with ERK and TFEB S142 phosphorylation in A375 cells. A375 and MeWo cells stably expressing TFEB-GFP were treated with PLX4720 (1 μ M, 12 hr), and WCLs were immunoprecipitated (IP) with anti-GFP-Trap beads, followed by immunoblotting (IB) with

3. S142 is also phosphorylated by mTORC1

in a Rag GTPase dependent manner. The authors show that a constitutively active mTOR (E2419K) does not prevent nuclear shuttling of TFEB in response to BRAFi treatment. Are the mTOR and ERK CA mutants expressed at comparable levels? The authors should use mTOR related perturbations that address the importance of Rag mediated activation in regulation of TFEB phosphorylation and localization. For instance - use a constitutively active RagGTPase (Q99), knockdown of GATOR1 components (renders RagA constitutively active).

Similarly Raptor knockdown does not appear to override constitutively active ERK in maintaining TFEB-GFP in the cytoplasm. In this experiment a western blot showing the efficiency of Raptor KD should be shown and quantification of nuclear vs cytoplasmic localization should be presented.

Response: Thank the Reviewer for this constructive comment. Current studies suggest that TFEB phosphorylation undergoes complex regulation by diverse upstream kinases in response to different environmental cues^{2,16}. The Rag GTPase, which senses lysosomal amino acids and activates mTORC1, regulates TFEB phosphorylation and nuclear translocation mainly in response to nutrient starvation and/or lysosomal stress¹⁷, while its roles in ERK-activated BRAF^{V600E} melanoma cells remained unknown. Our study demonstrated that in BRAF^{V600E} melanoma cells, ERK-mediated S142 phosphorylation of TFEB plays a dominant role in TFEB regulation independently of mTORC1. To further justify this per reviewer's suggestion, we examined the effect of constitutively active (CA) RagB (Q99L) expression and depletion of DEPDC5, a key subunit of the GATOR1 complex, both of which led to constitutive mTORC1 activation, on PLX4720-induced TFEB nuclear translocation. As shown in Fig. 2f (also attached for the reviewer's reference), expression of RagB (Q99L) or silencing of GATOR1 component function activated mTORC1 but failed to preclude PLX4720-driven TFEB nuclear translocation, as seen with mTORC1(CA). New data has been provided in Fig. 2f with the corresponding text revised accordingly to include this critical point.

Fig. 2f

Fig. 3e

Fig. 2f Representative images and quantification of nuclear localization of endogenous TFEB (green) in A375 cells stably expressing the constitutive active (CA) form of mTORC1 (E2419K), RagB GTPase (Q99L), or ERK (R67S/D321N), or expressing DEPDC5-specific shRNA, with or without the treatment of PLX4720 (1 μ M, 12 hr), or ERK inhibitor FR180204 (ERKi, 10 μ M, 24 hr). IB showed RagB^{Q99L} expression and DEPDC5 knockdown with the consequent mTORC1 activation as indicated by p70S6K and 4EBP1 phosphorylation. **Fig. 3e** Representative confocal images and quantification of cytoplasmic retention of WT TFEB and S142A and S211A TFEB mutants in A375 cells stably expressing ERK(CA) in the presence or absence of PLX4720 (1 μ M, 12 hr), Leptomycin B (LMB; 20 nM, 2 hr) or treated with Raptor-specific or CRM1-specific shRNA. Immunoblotting of WCL shows Raptor and CRM1 expression

Furthermore, silencing Raptor to inactivate mTORC1 function cannot override ERK(CA)-mediated cytoplasmic retention of TFEB (WT and S211A), though TFEB^{S142A} mutant translocated to the nucleus regardless of Raptor status. We have now provided western blot data to show the efficiency of Raptor KD along with quantification of TFEB cytoplasmic retention in Fig. 3e of the revised manuscript (also

attached for reviewer's reference). The main text has been revised accordingly.

4. The in vitro and in vivo tumorigenicity assays are all conducted using A375 cells overexpressing TFEB mutants, which is a weakness. In some instances the levels of overexpression are not equal for the individual mutants (see figure 7e, supplemental figure 5a). What would be the effect of overexpressing these same constructs in a control cells line (ei - MeWo -Braf WT cells) on tumor growth? Can the authors use CRISPR editing of the TFEB locus to introduce the point mutants and confirm a subset of their findings?

Response: Thank the Reviewer for this critical comment. Please kindly note that overexpression of constitutively active TFEB (i.e. TFEB^{S142A}) or constitutively inactive TFEB (i.e. TFEB^{S142E}) would bypass BRAF^{V600E}-mediated inhibition or BRAFi-mediated activation because TFEB is a downstream target of oncogenic BRAF^{V600E} signaling, Therefore, one would expect that overexpression of these TFEB constructs in MeWo cell line would exhibit similar trend across groups as seen in PLX4720-sensitive A375 cells and PLX4720-resistant A375^R cells (Fig. 5 and Fig. 7), although its associated tumorigenesis might be less aggressive than BRAF^{V600E} melanoma cells. Because TFEB inactivation and re-activation are patho-physiologically related to oncogenic BRAF^{V600E} and BRAFi treatment, respectively, we chose not to use MeWo cells but have used three independent melanoma cell lines,

Fig. S5 (c) Tumor volume of xenografts formed after subcutaneous injection of NOD/SCID mice with TFEB knockdown (TFEB sh) A375 cells reconstituted with vector, WT or mutant TFEB as indicated. Control shRNA (pGIPZ)-transfected cells were injected in parallel. Results are the mean volume \pm SD for 5-6 mice per group per time point. **(d)** Bioluminescence images (top) of tumor volume of the indicated A375 xenograft tumor genotype in live NOD/SCID mice at Day 28 after inoculation. Radiant efficiency expressed as p/sec/cm²/sr/(μ W/cm²) was quantified (bottom). **(e)** Colonogenic survival of TFEB knockdown A375 melanoma cells reconstituted with vector, WT or mutant TFEB as indicated. Representative images of colony-forming ability are shown (right panel); western blots show TFEB protein expression (bottom).

A375, A375^R, and B16, to demonstrate the gain/loss-of-TFEB function in BRAF^{V600E}-driven tumorigenesis and metastasis (Fig. 5-7). Nevertheless, the reviewer's concern regarding the overexpression system in tumorigenesis has been diligently considered. To address this, we employed an shRNA rescue strategy in which rescue constructs contain conservative point mutations that render them resistant to shRNA-mediated gene knockdown (KD). We generated TFEB-KD A375 cells and reconstituted these cells with WT TFEB or TFEB mutants to physiological levels (Fig. S5c-e; also refer to the attached figures). As expected, A375 cells expressing physiologically relevant levels of TFEB^{S142E} exhibited significantly increased clonogenicity in vitro and xenograft tumorigenesis in vivo, whereas the opposite was observed in cells expressing TFEB^{S142A} cells (Fig. S5c-e). Together, these results indicate that TFEB S142 phosphorylation and autophagy-lysosome suppression contribute to BRAF^{V600E}-driven tumorigenesis in melanoma cells.

5. The authors should more rigorously establish whether the effects of TFEB suppression on EMT and tumor growth are indeed autophagy/lysosome dependent or independent. The authors suggest that EMT induction and aggressive tumor growth following expression of inactive TFEB is due to a block in autophagy mediated degradation of TGFbeta. How is specificity for this cargo established? How relevant is this cargo relative to other proteins that are stabilized following TFEB inactivation? Wouldn't autophagy blockage via ATG knockdown, chloroquine treatment (independent of TFEB) have the same effect? Have the authors tested this? Do Braf mutant melanoma cells have overall higher TGFbeta

Fig. S7 (a,b) Western blot analysis of TGF- β and autophagy markers in A375 cells treated with chloroquine(CQ; 50 μ M) for indicated time (a) or in A375 cells with shRNA-mediated depletion of Atg5 and Beclin1 (b). Actin serves as a loading control. (c) Quantitative RT-PCR analysis of TGF- β mRNA expression in cells in (a) and (b). (e) Expression of TFEB^{S142A} promotes interaction between endogenous pro-TGF- β 1 and LC3 in A375 cells, which is inhibited by 3-MA (100 nM, 24 hr). (g) Western blot analysis of TGF- β and autophagy markers in BRAF^{WT} (MeWo) and BRAF^{V600E} mutant (A375 and G361) melanoma cells. Quantitative RT-PCR analysis of TGF- β mRNA expression in these cells is also shown (bottom).

Response: We thank the reviewer for this insightful comment. To further clarify the mechanism of TGF- β upregulation upon TFEB inactivation in melanoma cells, we rigorously conducted a series of new experiments on autophagy-lysosome-mediated TGF- β protein turnover and included the new datasets in Fig.S7a to S7h of the revised manuscript (also attached for reference). Briefly, the increase in TGF- β protein without a concomitant increase in mRNA level upon TFEB^{S142E} expression (or reversely upon TFEB^{S14A} expression) suggested that TGF- β protein is subjected to post-translational regulation. Indeed, we found that inhibition of autophagic flux by the lysosomotropic chloroquine (CQ), or silencing autophagy essential genes Beclin1 and Atg5, markedly increased TGF- β proteins in A375 cells without affecting the mRNA levels (Fig. S6a-c), indicating a steady-state turnover of TGF- β through the autophagy-lysosomal pathway. Notably, TGF- β protein contains potential LC3-interacting region (LIR) motifs (Fig. S7d), a core consensus sequence of (WFY)XX(ILV)⁷⁻⁹. We therefore tested whether LC3 binds to TGF- β targeted for selective turnover in the lysosomes. As expected, endogenous interaction between pro-TGF- β and LC3 was readily detected by co-immunoprecipitation in A375 cells even under basal condition (Fig. S7e). Expression of TFEB^{S142A} induced autophagy activation and concomitantly promoted pro-TGF- β association with LC3. Treatment of cells with 3-methyladenine (3-MA), a class III PI3K inhibitor that blocks autophagosome biogenesis, ablated the effect of TFEB^{S142A}, resulting in TGF- β accumulation (Fig. S7e). In accord, a significant quantity of pro-TGF- β was present in LC3-labeled autophagosomes and LAMP1-labeled lysosomes upon TFEB^{S142A} expression, whereas much less was found in TFEB^{S142E}-expressing cells (Fig. S7f). As speculated by the reviewer, the suppressed autophagy-lysosome activity in BRAF^{V600E} melanoma cells (A375 and G361) was associated with overall higher levels of TGF- β protein (not mRNA) as compared with that in MeWo cells (Fig. S7g). Furthermore, disruption of

cytoplasmic autophagy by depletion of Beclin1¹⁰ or inhibition of lysosomal function by Bafilomycin A1 (BafA1)¹¹ restored TGF- β protein levels that were suppressed by TFEB^{S142A} in A375 cells (Fig. S7h). Together, these results indicate that TGF- β is a selective cargo of autophagy and that TFEB acts on TGF- β protein turnover through the autophagy-lysosomal pathway. The new data has been included in Figure S7a to S7h of the revised manuscript with the corresponding text revised to further justify this critical point.

5. what are the baseline levels of autophagy, TFEB nuclear status, EMT in A375 versus A375R cells ? Are A375R cells more or less aggressive than the parental cell line?

Response: Thank the Reviewer for this interesting suggestion. A375^R melanoma cells carry a second NRAS(Q61K) mutation upstream of BRAF^{L8}, which activates ERK regardless of PLX4720 treatment. We have shown in the original manuscript (Fig. S8a; now Fig. S9a) that the basal autophagy function between A375 and A375^R melanoma cells are comparable, as indicated by p62 turnover and LC3 mobility shift (refer to the attached figures), and that TFEB is S142 phosphorylated and displayed cytoplasmic distribution in both cells (now Fig.S9a,b; refer to the attached figures). We have now provided new data of TGF- β and EMT protein expression. As shown in Fig. S9a, no significant difference was observed in TGF- β and EMT protein expression at basal levels between A375 and A375^R melanoma cells. However, in A375 cells, PLX4720 treatment induced autophagy-lysosome activation, diminished TFEB S142 phosphorylation (indicated by p-S142-TFEB specific antibody), translocated TFEB to the nucleus, increased E-cadherin levels, and decreased N-cadherin and Vimentin levels, which all correlated with a significant reduction of TGF- β protein. By striking contrast, A375^R melanoma cells showed no response to PLX4720 in all these relevant factors. The new data has been included in Figure S9a-b of the revised manuscript with the manuscript revised correspondingly.

Fig. Immunoblot analyses of indicated proteins in A375 and A375^R cells in the presence and absence of PLX4720 (1 μ M, 12 hr). Also refer to **Fig. S9a**.

Finally, the authors discuss that the role of autophagy in tumor growth is context dependent and potentially tissue specific – they support this idea with several references. However, their findings in melanoma contrast several reports in the same tissue type showing that autophagy activation is required for melanoma growth – eg. ATG5/7 knockout in Braf mutant melanoma mouse model suppresses tumor growth: Xie X et al Cancer Discov 2015; hydroxychloroquine is associated with tumor inhibition in melanoma patient clinical trials: Rangwala R et al Autophagy 2014; lysosomal inhibition inhibits melanoma growth: Rebecca VW et al Cancer Discov 2017; patients with autophagy induction following Braf inhibitor treatment showed reduced response to therapy and poor prognosis: Ma et al J Clin Invest. These studies all suggest that autophagy activation is required for tumor growth and portend worse overall outcomes. How do the authors reconcile their data with these previously published findings in the same cancer type?

Response: We appreciate the Reviewer's comment on this controversial topic and on the paradoxical role of autophagy in both suppressing and promoting cancer development/progression in different settings. Our results differed from the work of Xie et al., who reported that melanocyte-specific Atg7 deletion prevented, rather than promoted, melanomagenesis by BRAF^{V600E} and allelic Pten loss in a mouse model¹⁹. This paradox is unresolved but could be rationalized through the view of a differential role of autophagy null vs. autophagy suppression in distinct stages of tumor development. Despite the

general consensus that suppressed autophagy, as exemplified by monoallelic loss of autophagy essential genes such as *Becn1*, favors tumor initiation¹⁵, it is possible that autophagy null due to genetic *Atg7* knockout can ablate basal cell fitness and cause growth defects of melanocytes even in the presence of *BRAF*^{V600E}, thereby leading to reduced melanocyte transformation (pigmented lesion) and melanomagenesis. In fact, the stunted melanoma growth in the work of Xie et al., can only be observed upon complete *Atg7* knockout (autophagy null) but not upon monoallelic *Atg7* knockdown (autophagy suppression). It is also worth noting that the autophagy-lysosome pathway is not an isolated process from oncogenic programming, rather, it is directly regulated by oncogenic signaling¹⁵. Therefore, to decipher the relative contribution of autophagy to oncogene-driven melanoma, it is always important to first understand the mechanisms of autophagy regulation/dysregulation by oncogenic signaling or related targeted therapy. In our study, we focused on the molecular mechanisms by which oncogenic *BRAF*^{V600E} and its inhibitors regulate autophagy at the transcriptional level through ERK-mediated TFEB phosphorylation/inactivation. To further determine whether this regulation contributes to *BRAF*^{V600E}-driven oncogenesis vs. a bystander event, we used *TFEB*^{S142A} to revert the effect of *BRAF*^{V600E} on TFEB inhibition and found it actually inhibited (rather than promoted) *BRAF*^{V600E}-driven melanoma progression. Thus, our findings demonstrate that at least in this patho-physiologically relevant context, activation of autophagy-lysosome function by TFEB contributes to inhibition (rather than promotion) of *BRAF*^{V600E}-associated tumor progression. Furthermore, Xie et al.¹⁹, used 4-HT to simultaneously induce *BRAF*^{V600E} overexpression and *Pten* loss in parallel with *Atg7* deletion in normal melanocytes and then examine their neoplastic transformation (pigmented lesion) and progression into melanoma tumors. We are investigating how TFEB (activation vs. inactivation)-mediated autophagy-lysosome function influences progression of established tumors and/or their response to therapy. Although both *Atg7* ablation or TFEB inactivation increased mitochondria damage and ROS production in cells, such increased oxidative stress might be detrimental to overall survival of primary melanocytes during transformation but more tolerated in established tumor cells due to induction of genomic instability. Moreover, our RNA-seq analysis, cell-based assay, and xenograft models confirmed the aberrant upregulation of TGF- β signaling associated with impaired autophagy-lysosome function, thus adding a new mechanism of tumor progression. We have further discussed this issue in the revised manuscript.

As discussed in the original manuscript, the current belief that autophagy inhibition may promote tumor inhibition^{20,21} is largely based on studies of anti-tumor effects of the lysosomotropic antimalarial drug chloroquine or its derivatives in combination cancer chemotherapy^{14,21,22}, as exemplified in Rangwala R et al, (*Autophagy*2014), Rebecca VW et al., (*Cancer Discov.* 2017), and Ma et al. (*JCI*, 2014). We would like to respectfully point out that CQ, which affects all acidic compartments in cells, is not a selective autophagy inhibitor, and that CQ has anti-tumor activities beyond autophagy and even lysosomal functions (such as DNA intercalation)^{15,23}. For instance, CQ is recently reported to switch tumor-associated macrophages from M2 toward tumor-killing M1 phenotype through the activation of p38 and NF- κ B (Chen et al., *Nat Commun.* 2018)²³. It is also worth noting that CQ treatment alone is sufficient to induce TFEB nuclear translocation and activation of lysosomal and autophagy genes^{1,24}, which may contribute to tumor regression in combination therapy. Thus, the mechanistic underpinnings of CQ-based trials await further investigation.

Finally, to manipulate the BRAFi-induced autophagy upregulation, it is important to first understand the molecular mechanisms by which BRAFi regulates autophagy. Ma et al.²¹ reported that BRAFi induces autophagy through a MAPK-driven expansion of the ER stress response. However, our rigorous and repeated experiments indicate that blunting ER stress cannot forestall BRAF inhibitor-induced autophagy upregulation. We further found that BRAFi-induced autophagy is not only a cytoplasmic process but more a transcriptional event through TFEB, and that inactivation of TFEB abrogated BRAFi activity (refer to Fig. 1 and S1). By using comprehensive biochemical, molecular

biological, and cell-based assay we confirmed the TFEB-dependent mechanism of BRAFi in autophagy promotion. Thus, instead of using non-selective CQ, we employed a mechanism-driven strategy by expressing a TFEB phosphomimetic to revert the effect of PLX4720 on autophagy-lysosome function for both in vitro and in vivo assays. In all these studies, autophagy-lysosomal inhibition due to TFEB inactivation was associated with increased clonogenic survival and BRAFi resistance, accompanied by increased tumor dissemination in xenograft in vivo. Unlike many other studies that use tumor shrinkage as the only measure of treatment efficacy while neglecting other progression-related histopathological changes, we further identified increased TGF- β and its signaling upregulation may contribute to the aggressive and resistant phenotype upon blockade of BRAFi-induced autophagy-lysosome activation. The data presented here did not address the potential beneficial effects of autophagy inhibition on melanoma growth. Rather, the data indicated that autophagy-lysosome induction may contribute to BRAFi response in melanoma with active BRAF^{V600E}; and that the use of proximal inhibitors of autophagy may adversely (rather than favorably) affect their clinical outcome.

We have discussed in-depth these discrepancies in the revised manuscript and believe this mechanism-driven new finding may provide important insights into the complex role of autophagy-lysosome function in different tumor settings.

Minor points:

1. The autoradiograph in figure 3d related to the kinase assay is not convincing and should be replaced.

Response: *The in vitro phosphorylation assay in Fig. 3d has been repeated per reviewer's suggestion. Consistent results were achieved and new data of improved quality has been provided in the revised manuscript.*

2. p-ERK levels should be shown in Fig S9d

Response: *We thank the reviewer for this comment. Western blot data of p-ERK and total-ERK has now been included in Fig. S10d (original Fig. S9d) of the revised manuscript, whereby PLX4720 treatment inhibited ERK activation.*

3. the concentration of PLX4720 used and duration of treatment should be indicated in the figure legend for each experiment (eg. missing in fig S9c)

Response: *We apologize for this mistake. The concentration of PLX4720 and treatment time has now been included in all figures and supplementary figures in the revised manuscript.*

Again, we thank the Reviewer for his/her suggestions that have led to a much-improved manuscript.

Reference:

- 1 Rocznik-Ferguson, A. *et al.* The transcription factor TFEB links mTORC1 signaling to transcriptional control of lysosome homeostasis. *Sci Signal* **5**, ra42, doi:10.1126/scisignal.2002790 (2012).
- 2 Napolitano, G. & Ballabio, A. TFEB at a glance. *J Cell Sci* **129**, 2475-2481, doi:10.1242/jcs.146365 (2016).
- 3 Mizushima, N. & Yoshimori, T. How to interpret LC3 immunoblotting. *Autophagy* **3**, 542-545 (2007).
- 4 Rogakou, E. P., Pilch, D. R., Orr, A. H., Ivanova, V. S. & Bonner, W. M. DNA double-stranded breaks induce histone H2AX phosphorylation on serine 139. *J Biol Chem* **273**, 5858-5868 (1998).
- 5 Marino, G., Niso-Santano, M., Baehrecke, E. H. & Kroemer, G. Self-consumption: the interplay of autophagy and apoptosis. *Nat Rev Mol Cell Biol* **15**, 81-94, doi:10.1038/nrm3735 (2014).
- 6 Li, M. *et al.* The ATM-p53 pathway suppresses aneuploidy-induced tumorigenesis. *Proc Natl Acad Sci U S A* **107**, 14188-14193, doi:10.1073/pnas.1005960107 (2010).
- 7 Pankiv, S. *et al.* p62/SQSTM1 binds directly to Atg8/LC3 to facilitate degradation of ubiquitinated protein aggregates by autophagy. *J Biol Chem* **282**, 24131-24145, doi:10.1074/jbc.M702824200 (2007).
- 8 Noda, N. N. *et al.* Structural basis of target recognition by Atg8/LC3 during selective autophagy. *Genes Cells* **13**, 1211-1218, doi:10.1111/j.1365-2443.2008.01238.x (2008).
- 9 Jacomin, A. C., Samavedam, S., Promponas, V. & Nezis, I. P. iLIR database: A web resource for LIR motif-containing proteins in eukaryotes. *Autophagy* **12**, 1945-1953, doi:10.1080/15548627.2016.1207016 (2016).
- 10 Liang, X. H. *et al.* Induction of autophagy and inhibition of tumorigenesis by beclin 1. *Nature* **402**, 672-676 (1999).
- 11 Yoshimori, T., Yamamoto, A., Moriyama, Y., Futai, M. & Tashiro, Y. Bafilomycin A1, a specific inhibitor of vacuolar-type H(+)-ATPase, inhibits acidification and protein degradation in lysosomes of cultured cells. *J Biol Chem* **266**, 17707-17712 (1991).
- 12 Kuzu, O. F., Nguyen, F. D., Noory, M. A. & Sharma, A. Current State of Animal (Mouse) Modeling in Melanoma Research. *Cancer Growth Metastasis* **8**, 81-94, doi:10.4137/CGM.S21214 (2015).
- 13 Fidler, I. J. The relationship of embolic homogeneity, number, size and viability to the incidence of experimental metastasis. *Eur J Cancer* **9**, 223-227 (1973).
- 14 Rebecca, V. W. *et al.* A Unified Approach to Targeting the Lysosome's Degradative and Growth Signaling Roles. *Cancer Discov* **7**, 1266-1283, doi:10.1158/2159-8290.CD-17-0741 (2017).
- 15 Rubinsztein, D. C., Codogno, P. & Levine, B. Autophagy modulation as a potential therapeutic target for diverse diseases. *Nat Rev Drug Discov* **11**, 709-730, doi:10.1038/nrd3802 (2012).
- 16 Settembre, C. & Medina, D. L. TFEB and the CLEAR network. *Methods Cell Biol* **126**, 45-62, doi:10.1016/bs.mcb.2014.11.011 (2015).
- 17 Settembre, C. *et al.* A lysosome-to-nucleus signalling mechanism senses and regulates the lysosome via mTOR and TFEB. *EMBO J* **31**, 1095-1108, doi:10.1038/emboj.2012.32 (2012).
- 18 Nazarian, R. *et al.* Melanomas acquire resistance to B-RAF(V600E) inhibition by RTK or N-RAS upregulation. *Nature* **468**, 973-977, doi:10.1038/nature09626 (2010).
- 19 Xie, X., Koh, J. Y., Price, S., White, E. & Mehnert, J. M. Atg7 Overcomes Senescence and Promotes Growth of BrafV600E-Driven Melanoma. *Cancer Discov* **5**, 410-423, doi:10.1158/2159-8290.CD-14-1473 (2015).
- 20 Amaravadi, R. K. *et al.* Principles and current strategies for targeting autophagy for cancer treatment. *Clin Cancer Res* **17**, 654-666, doi:10.1158/1078-0432.CCR-10-2634 (2011).
- 21 Ma, X. H. *et al.* Targeting ER stress-induced autophagy overcomes BRAF inhibitor resistance in melanoma. *J Clin Invest* **124**, 1406-1417, doi:10.1172/JCI70454 (2014).
- 22 Rangwala, R. *et al.* Combined MTOR and autophagy inhibition: phase I trial of hydroxychloroquine and temsirolimus in patients with advanced solid tumors and melanoma. *Autophagy* **10**, 1391-1402, doi:10.4161/auto.29119 (2014).
- 23 Chen, D. *et al.* Chloroquine modulates antitumor immune response by resetting tumor-associated macrophages toward M1 phenotype. *Nat Commun* **9**, 873, doi:10.1038/s41467-018-03225-9 (2018).
- 24 Emanuel, R. *et al.* Induction of lysosomal biogenesis in atherosclerotic macrophages can rescue lipid-induced lysosomal dysfunction and downstream sequelae. *Arterioscler Thromb Vasc Biol* **34**, 1942-1952, doi:10.1161/ATVBAHA.114.303342 (2014).

Reviewers' comments:

Reviewer #1 (Remarks to the Author):

After careful examination I believe the authors have completely addressed any concern of my previous review. They have also significantly improved the manuscript.

I would therefore recommend this manuscript for publication

Reviewer #3 (Remarks to the Author):

The authors have provided additional data that address most of the comments brought up during the initial round of revision.

A remaining concern relates to how the authors justify selective targeting of TGFb for degradation via autophagy. The authors describe a putative LIR domain in TGFb however this consensus does not strictly conform with parameters known to be important for canonical LIRs such as that in p62 which contains acidic residues N terminal to the Tryptophan (most commonly aspartic acid). Also the immunoprecipitation in suppl fig 7e is not well controlled - no IgG control, TGFb-LIR mutant is used to support the authors claim that TGFb directly binds to LC3. Moreover this direct binding seems quite unusual as most selective autophagy cargos are associated with posttranslational modifications (such as ub) which is recognized by an adaptor (eg. p62) containing both a LIR and a ub binding region which subsequently bridges cargo destined for degradation with LC3.

Related to this point the images in Suppl figure 7f of TGFb, LC3 and lamp are not convincing. The overlap does not appear to be specific at the resolution shown, while the lack of colocalization with lamp in the context of the S142E mutant are likely due to the poor lamp staining in the selected images. Overall while these data attempt to address my question relating to specificity for TGFb degradation the data as presented are not convincing.

Finally I remain concerned about the use of overexpression constructs of TFEB in this study. While the authors have conducted some of these experiments in the context of TFEB knockdown, the expression of exogenous mutants is still not physiological as they are expressed at a substantially higher level compared to endogenous TFEB in these cells. Thus the slower growth of cells/tumors following overexpression of TFEB S142A (nuclear) may simply be due to a heightened level of general catabolic activity associated with increased autophagy/lysosome induction and not due to specific degradation of TGFb. Along these lines it is likely that TFEB S142A would slow growth in any cell line when overexpressed above physiological levels - hence my suggestion to perform this experiment in control cell lines in point 4 – which was not done.

RESPONSE TO REVIEWERS

We thank both reviewers for their thoughtful comments on our revised manuscript. As Reviewer 1 was completely satisfied with the revised manuscript, this letter only focuses on addressing the remaining concerns raised by Reviewer 3. In this revised manuscript, we have added new EXPERIMENTAL data, hopefully adequately addressing the Reviewer's comments. We hope the new version of our manuscript is now appropriately suited for publication in the *Nature Communications* Journal.

Point-by-point Response to Reviewer #3

Reviewer 3 stated that the authors have provided additional data that address most of the comments brought up during the initial round of revision. We are truly appreciative of the reviewer 3's comments. Rigorous efforts have been taken to address the Reviewer's comments point-by-point. The new data are included in Supplemental Figure 7d-h with relevant text revised to reflect changes. A detailed response to the Reviewer's critiques and a description of the new experiments (in italics) follow:

1. A remaining concern relates to how the authors justify selective targeting of TGF β for degradation via autophagy. the authors describe a putative LIR domain in TGF β however this consensus does not strictly conform with parameters known to be important for canonical LIRs such as that in p62 which contains acidic residues N terminal to the Tryptophan (most commonly aspartic acid). Also the immunoprecipitation in suppl fig 7e is not well controlled - no IgG control, TGF β -LIR mutant is used to support the authors claim that TGF β directly binds to LC3. Moreover this direct binding seems quite unusual as most selective autophagy cargos are associated with posttranslational modifications (such as ub) which is recognized by an adaptor (eg. p62) containing both a LIR and a ub binding region which subsequently bridges cargo destined for degradation with LC3.

Response: *The reviewer raises a good point and we thank the Reviewer for this comment. Although the conventional LIR sequence was initially described as the **W-x-x-L** motif, where the acidic residues N-terminal to the Tryptophan was found important for p62 interaction with LC3 (Ichimura et al., 2008; Noda et al., 2008; Pankiv et al., 2007), a recent study of experimentally verified LC3-interactors has redefined the sequence of LIR to xLIR motif - (ADEFGLPRSK)(DEGMSTV)(WFY)(DEILQTV)(ADEFHIKLM PSTV)(ILV) (Jacomin et al., 2016). Using the established iLIR database (<https://ilir.warwick.ac.uk>), we were able to identify two putative*

Fig. S7g Pulldown (PD) assays using recombinant His-LC3 and lysates from A375 cells expressing Flag-tagged WT pro-TGF- β or W17A/V20A, F257A/L260A, W17A/V20A/F257A/L260A pro-TGF- β mutants. Note significantly reduced LC3 binding with pro-TGF- β mutants. Actin serves as a loading control. Input recombinant proteins visualized with Coomassie blue staining are shown (lower panel). **Fig. S7h** Recombinant His-LC3 interacts with recombinant GST-tagged WT pro-TGF- β but not its W17A/V20A/F257A/L260A mutant. Lower panels show input recombinant proteins by Coomassie blue staining.

LC3-binding regions in TGF- β (Fig. S7d). To experimentally verify whether the xLIR motifs of TGF- β mediate LC3 interaction, we generated the W17A/V20A (motif 1), F257A/L260A (motif 2), and W17A/V20A/F257A/L260A (combined 1&2) point mutants of TGF- β (Fig. S7g). Pull-down assays using recombinant His-LC3 and lysates from cells expressing Flag-tagged wild-type TGF- β or its xLIR-mutants revealed that the TGF- β -LC3 interaction was strongly reduced by W17A/V20A or F257A/L260A (Lane 5 and Lane 6, Fig. S7g), and was almost abolished by the W17A/V20A/F257A/L260A mutant (Lane 4, Fig. S7g), suggesting that both xLIR motifs are required for efficient LC3 interaction. We further confirmed their direct binding by using purified recombinant GST-TGF- β and His-LC3 in a cell-free environment, which again was abolished by W17A/V20A/F257A/L260A (Fig. S7h). These results, together with the data of endogenous TGF- β -LC3 interaction (Fig. S7e) and their subcellular colocalization in vivo (Fig. S7f), strongly support that the autophagy protein LC3 directly interacts with TGF- β and selectively targets TGF- β for autophagic degradation.

It is noteworthy that although p62 is an important autophagy adaptor, p62- and by extension adaptors-independent cargo selection and autophagy regulation have been revealed in recent studies. For instance, AMBRA1, which also contains a non-canonical LIR (SGVEYYWxxL), directly interacts with LC3 and regulates mitophagy (Strappazzon et al., 2015). Moreover, the putative LIR motif (PPSHWxxL) of β -Catenin was found to be required for its direct interaction with LC3 and autophagy-mediated degradation (Petherick et al., 2013). Furthermore, given the extensive web of LC3 interactome (Wild et al., 2014), it is not surprising to identify TGF- β as another selective cargo for autophagic turnover. Rather, the crosstalk between TGF- β signaling and the autophagy pathway, identified in our study, provides novel insights into the role of autophagy in cancer.

Finally, Fig. S7e has been repeated with the control IgG included and consistent results achieved. We have attached (see above) Fig. S7g and S7h for your reference.

2. Related to this point the images in Suppl figure 7f of TGF β , LC3 and lamp are not convincing. The overlap does not appear to be specific at the resolution shown, while the lack of colocalization with lamp in the context of the S142E mutant are likely due to the poor lamp staining in the selected images. Overall while these data attempt to address my question relating to specificity for TGF β degradation the data as presented are not convincing.

Response: We thank Reviewer for this comment. We have rigorously re-conducted confocal microscopy analysis on the relative distribution of TGF- β to the LAMP1-labelled lysosome and LC3-labelled autophagosome with improved resolution and high consistency. The new data are included in Fig. S7f (attached to the right). As indicated in our repeated experiments, a significant quantity of TGF- β was present in LC3-labeled autophagosomes and LAMP1-labeled lysosomes in cells expressing TFEB^{S142A}, whereas much less was detected upon TFEB^{S142E} expression (Fig. S7f). Quantification of relative colocalization of TGF- β with LAMP1 or LC3 in approximately

Figure S7f

Fig. S7f. Subcellular distribution of TGF- β (green) relative to the lysosomes (red) or to the LC3-labelled autophagosomes (red) in A375 cells expressing vector, TFEB^{S142A} or TFEB^{S142E}. The percentage of TGF- β associated with the autophagy and lysosomal compartments are quantified (right panels).

200 cells, randomly chosen from 10 high power fields, further consolidate our conclusion of autophagy-dependent degradation of TGF- β , which is promoted by TFEB^{S142A} but inhibited by TFEB^{S142E}.

Regarding the poor LAMP staining in TFEB^{S142E} cells as noted by the reviewer, we would like to respectfully point out that the overall staining of autophagosome (indicated by LC3) and lysosomes (indicated by LAMP1) in TFEB^{S142E} cells is indeed lower due to expression of the constitutively inactive mutant TFEB, whereas it is higher in TFEB^{S142A} cells due to expression of the constitutively active mutant TFEB, compared to the vector control cells. Conversely, the levels of TGF- β is inversely correlated with the levels of autophagy-lysosomal activity. This is a consistent phenotype throughout the manuscript. Despite the difference in their expression levels, the relative distribution pattern of TGF- β /LC3 or TGF- β /LAMP1 in TFEB^{S142E} vs. TFEB^{S142A} cells is clearly different. In TFEB^{S142A} cells, TGF- β is largely enriched in the area co-stained with LC3 or LAMP1, whereas in TFEB^{S142E} cells, the TGF- β -enriched area is largely devoid of LC3 or LAMP1 staining. Please kindly note that all images between different cell lines were captured in parallel using the same confocal parameters. Although the LAMP1 staining in TFEB^{S142E} cells can be further improved with increased resolution or visualization, this will result in signal oversaturation in TFEB^{S142A} cells under the same setting. Nevertheless, given our highly replicable results and biochemical support of TGF- β interaction with the autophagy-lysosome pathway, we hope the reviewer will now find the data satisfactory.

3. Finally I remain concerned about the use of overexpression constructs of TFEB in this study. While the authors have conducted some of these experiments in the context of TFEB knockdown, the expression of exogenous mutants is still not physiological as they are expressed at a substantially higher level compared to endogenous TFEB in these cells. Thus the slower growth of cells/tumors following overexpression of TFEB S142A (nuclear) may simply be due to a heightened level of general catabolic activity associated with increased autophagy/lysosome induction and not due to specific degradation of TGF β . Along these lines it is likely that TFEB S142A would slow growth in any cell line when overexpressed above physiological levels - hence my suggestion to perform this experiment in control cell lines in point 4 – which was not done.

Response: To address the reviewer's concern regarding the overexpression system, we have employed a shRNA rescue strategy in which rescue constructs contain conservative point mutations that render them resistant to shRNA-mediated gene knockdown (KD). As shown in original Fig. S5e in the revised manuscript (also attached here for reference), the expression of exogenous (reconstituted) WT TFEB or TFEB mutants (Lane 3, 4, 5) in TFEB-knockdown cells is comparable to endogenous ones in control shRNA-treated cells (Lane 1). In other words, both the WT and mutant TFEB expression in this system is physiologically relevant. Using this system, we had been able to justify that cells expressing physiologically relevant levels of TFEB^{S142E} exhibited significantly increased clonogenicity in vitro and xenograft tumorigenesis in vivo, whereas the opposite was observed in cells expressing TFEB^{S142A} cells (Fig. S5c-e). Therefore, we are confused by the reviewer's comment on the "substantially higher level compared to endogenous TFEB". Nevertheless, both in vitro and in vivo studies unambiguously demonstrate that BRAF^{V600E}-associated TFEB S142 phosphorylation and ensuing autophagy-lysosome suppression contribute to BRAF^{V600E}-driven tumorigenesis in melanoma cells.

Secondly, we would like to further reason why BRAF mutant rather than BRAF WT cell lines should be used in this study. We would like to kindly remind that one goal of this work is to elucidate the significance of (BRAF^{V600E}-ERK)-mediated TFEB phosphorylation at S142 in BRAF^{V600E}-driven

melanomagenesis and in BRAFi therapy, and that TFEB S142 phosphorylation is a dominant event only in BRAF^{V600E} cells but not in BRAF WT cells. Thus, it is biologically more relevant to investigate the impact of the constitutive expression of TFEB^{S142E} (which enhanced BRAF^{V600E} inhibition of TFEB) and of TFEB^{S142A} (which abrogated BRAF^{V600E} inhibition of TFEB) in melanoma cells with active BRAF mutation, rather than in BRAF WT cells. On the other hand, given the general tumor-suppressing property of the autophagy-lysosomal pathway and its transcriptional upregulation by TFEB, it is not surprising (actually highly expected) that expression of TFEB^{S142A} will also slow tumor growth in BRAF WT cells. However, regardless of whether TFEB S142 phosphorylation functions similarly or differently in WT vs. BRAF^{V600E} cells, we believe it will not provide further insights into the interrelationship between oncogenic signaling and TFEB-mediated transcriptional regulation of the autophagy-lysosome pathway. Therefore, we remain to use BRAF mutant cells to position the newly identified mechanisms of TFEB regulation in the biologically and clinically relevant context.

Finally, we thank the Reviewer for all suggestions that have led to a much-improved manuscript, presented here for consideration.

Thank you.

REFERENCE

- Arasaki, K., Taniguchi, M., Tani, K., and Tagaya, M. (2006). RINT-1 regulates the localization and entry of ZW10 to the syntaxin 18 complex. *Mol Biol Cell* *17*, 2780-2788.
- Ichimura, Y., Kominami, E., Tanaka, K., and Komatsu, M. (2008). Selective turnover of p62/A170/SQSTM1 by autophagy. *Autophagy* *4*, 1063-1066.
- Jacomin, A.C., Samavedam, S., Promponas, V., and Nezis, I.P. (2016). iLIR database: A web resource for LIR motif-containing proteins in eukaryotes. *Autophagy* *12*, 1945-1953.
- Noda, N.N., Kumeta, H., Nakatogawa, H., Satoo, K., Adachi, W., Ishii, J., Fujioka, Y., Ohsumi, Y., and Inagaki, F. (2008). Structural basis of target recognition by Atg8/LC3 during selective autophagy. *Genes Cells* *13*, 1211-1218.
- Pankiv, S., Clausen, T.H., Lamark, T., Brech, A., Bruun, J.A., Outzen, H., Overvatn, A., Bjorkoy, G., and Johansen, T. (2007). p62/SQSTM1 binds directly to Atg8/LC3 to facilitate degradation of ubiquitinated protein aggregates by autophagy. *J Biol Chem* *282*, 24131-24145.
- Petherick, K.J., Williams, A.C., Lane, J.D., Ordonez-Moran, P., Huelsken, J., Collard, T.J., Smartt, H.J., Batson, J., Malik, K., Paraskeva, C., *et al.* (2013). Autolysosomal beta-catenin degradation regulates Wnt-autophagy-p62 crosstalk. *EMBO J* *32*, 1903-1916.
- Schweizer, A., Fransen, J.A., Matter, K., Kreis, T.E., Ginsel, L., and Hauri, H.P. (1990). Identification of an intermediate compartment involved in protein transport from endoplasmic reticulum to Golgi apparatus. *European journal of cell biology* *53*, 185-196.
- Strappazon, F., Nazio, F., Corrado, M., Cianfanelli, V., Romagnoli, A., Fimia, G.M., Campello, S., Nardacci, R., Piacentini, M., Campanella, M., *et al.* (2015). AMBRA1 is able to induce mitophagy via LC3 binding, regardless of PARKIN and p62/SQSTM1. *Cell Death Differ* *22*, 517.
- Wild, P., McEwan, D.G., and Dikic, I. (2014). The LC3 interactome at a glance. *J Cell Sci* *127*, 3-9.

REVIEWERS' COMMENTS:

Reviewer #3 (Remarks to the Author):

Based on the authors efforts to address my comments, I recommend acceptance of the manuscript